# JADE: Expert-Grounded Dynamic Evaluation for Open-Ended Professional Tasks

Lanbo Lin [* 1]  Jiayao Liu [* 1]  Tianyuan Yang [* 1]  Li Cai [1 2]  Yuanwu Xu [1]  Lei Wei [1 3]  Sicong Xie [1]
Guannan Zhang [1]

## Abstract

Evaluating agentic AI on open-ended professional tasks faces a fundamental dilemma between rigor and flexibility. Static rubrics provide rigorous, reproducible assessment but fail to accommodate diverse valid response strategies, while LLM-as-a-judge approaches adapt to individual responses yet suffer from instability and bias. Human experts address this dilemma by combining domain-grounded principles with dynamic, claim-level assessment. Inspired by this process, we propose **JADE**, a two-layer evaluation framework. Layer 1 encodes expert knowledge as a predefined set of evaluation skills, providing stable evaluation criteria. Layer 2 performs report-specific, claim-level evaluation to flexibly assess diverse reasoning strategies, with evidence-dependency gating to invalidate conclusions built on refuted claims. Experiments on BizBench show that JADE improves evaluation stability and reveals critical agent failure modes missed by holistic LLM-based evaluators. We further demonstrate strong alignment with expert-authored rubrics and effective transfer to HealthBench and DR.BENCH, covering medical and 10-domain professional evaluation settings. Code and data are available at https://github.com/smiling-world/JADE.

## 1. Introduction

The evolution of Large Language Models (LLMs) from chat-bots to reasoning engines has catalyzed the rise of autonomous agentic systems (Naveed et al., 2025; Wang et al., 2024a). By integrating multi-step reasoning with tool-use capabilities, these agents are increasingly deployed in complex, long-horizon professional workflows (Luo et al., 2025; Jimenez et al., 2024). As capabilities advance, robust evaluation (Yehudai et al., 2025; Du et al., 2026) becomes critical for guiding research and ensuring reliability in high-stakes applications.

Most existing agent benchmarks focus on tasks with verifiable outcomes, such as multi-hop web search (Wei et al., 2025; Zhou et al., 2025b; Chen et al., 2025b), code generation (Jimenez et al., 2024; Quan et al., 2025), or tool-mediated reasoning (Yao et al., 2025a; Wang et al., 2026; Barres et al., 2025), enabling objective and reproducible evaluation. However, such proxy tasks capture only a limited subset of real-world deployments.

In real-world applications, agents must solve open-ended problems involving strategic analysis, evidence synthesis, and decision-making under evolving conditions (e.g., strategic procurement, market analysis, and professional consulting). These tasks rarely admit a single "gold" answer, making realistic evaluation an unresolved challenge. For example, a query for "*FDA-certified suppliers for stainless steel tumblers with low shipping cost to the USA*" admits multiple valid response strategies, whose quality cannot be reliably assessed without consistent professional principles and real-time evidence verification.

Recent benchmarks (Du et al., 2026; Yao et al., 2025b) explore expert-authored rubrics and LLM-generated checklists to approximate professional judgment. While promising, these approaches expose a fundamental dilemma: (i) **Static checklists** ensure stability but lack adaptivity to diverse valid solutions (Arora et al., 2025; Ruan et al., 2026; Zhu et al., 2025); (ii) **LLM-as-a-judge methods** (Zheng et al., 2023; Du et al., 2026) provide adaptivity through direct scoring or LLM-generated query-specific checklists, but suffer from stochastic variance and systemic biases. More fundamentally, they lack domain-calibrated, expert-grounded evaluation principles and fail to construct explicit report-specific, claim-grounded checklists, hindering systematic assessment of evidential credibility and reasoning quality. We term this the **stability–adaptivity dilemma**.

Human experts resolve this challenge by decoupling domain-general evaluation principles from case-specific evidence,

---

[*]Equal contribution [1]Alibaba International Digital Commerce Group [2]Zhejiang University [3]Peking University. Correspondence to: Lanbo Lin <linlanbo.llb@alibaba-inc.com>.

*Proceedings of the 43rd International Conference on Machine Learning*, Seoul, South Korea. PMLR 306, 2026. Copyright 2026 by the author(s).

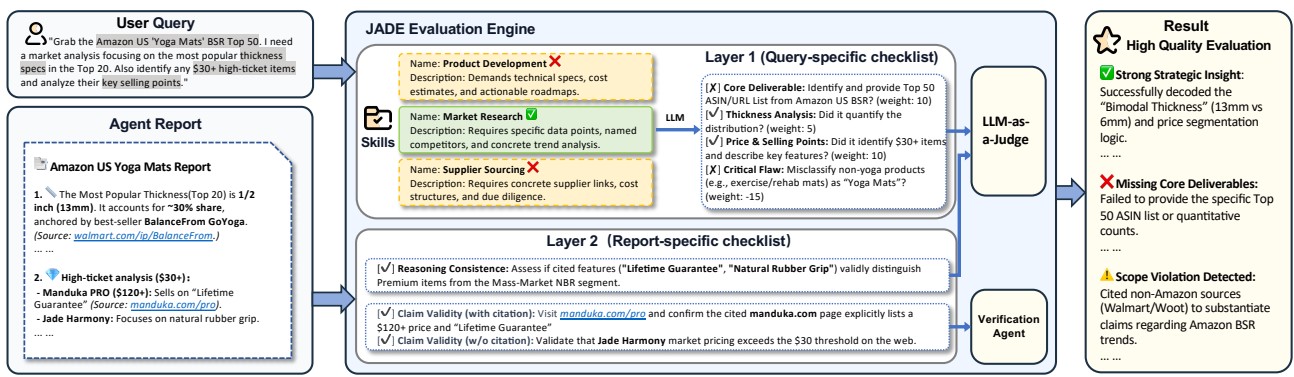

*Figure 1.* Example of End-to-End Evaluation of JADE on BizBench

rather than matching against predefined ideal answers. This insight motivates **J**udge **A**gent with **D**ynamic **E**valuation (JADE), which achieves both stability and adaptivity through a two-layer design: deterministic skill activation and adaptive claim-level verification. Overall, these components enable systematic, transparent, and context-aware evaluation. Our main contributions are:

- We propose JADE, a two-layer framework for stable and adaptive evaluation that encodes expert knowledge as reusable skills and performs dynamic, claim-level verification.

- We introduce BizBench, a benchmark of 150 labeled strategic sourcing queries for evaluating temporally dynamic professional workflows.

- Through JADE's structured evaluation on BizBench, we empirically identify systematic agent failure modes—a large evidence–reasoning gap and pervasive citation laundering—that are often missed by holistic LLM evaluators.

**Conflict of Interest Disclosure.** All authors are affiliated with Alibaba International Digital Commerce Group. The paper evaluates Qwen3-Max, which is developed by an Alibaba Group affiliate; the authors were not involved in developing Qwen3-Max.

## 2. Related Work

### 2.1. Deep Research Agents and Professional Benchmarks

Recent advances in LLMs have enabled *Deep Research Agents* (DRAs)(OpenAI, 2025a; Perplexity Team, 2025; xAI, 2025; Team et al., 2025; Google, 2025a) capable of multi-step information synthesis and tool-mediated reasoning in complex workflows. Correspondingly, a growing body of benchmarks has been proposed in domains such as academia (Zhou et al., 2025a; Liu et al., 2025; Wan et al.,

2026), healthcare (Arora et al., 2025), finance (Hu et al., 2026; Sun et al., 2025; Zhu et al., 2025; Chen et al., 2025a), and e-commerce (Yao et al., 2022; Wang et al., 2026; 2025; Min et al., 2025). While these benchmarks capture important aspects of agentic reasoning, most emphasize task completion or short-horizon accuracy, leaving systematic evaluation of long-form professional reports underexplored in both realistic benchmark design and stable yet adaptive evaluation methodology.

### 2.2. Benchmark Environments and Data Construction

The realism of agentic benchmarks depends on both environmental dynamics and data fidelity.

**Environment Dynamics.** Early benchmarks (Zhou et al., 2024; Yao et al., 2022; 2025a; Wei et al., 2024) rely on sandboxed environments or static snapshots, enabling reproducibility but failing to capture temporal variability. Recent work advocates Live Web evaluation (Zeng et al., 2026; Hu et al., 2026; Wang et al., 2025), which BizBench follows in strategic sourcing scenarios.

**Data Fidelity.** Benchmarks such as BrowseComp (Wei et al., 2025) employ synthetically constructed queries that may deviate from natural user distributions, whereas Health-Bench (Arora et al., 2025) and AssistantBench (Yoran et al., 2024) emphasize authentic user needs. BizBench prioritizes naturally occurring professional inquiries to reflect realistic decision-making constraints.

### 2.3. Evaluation of Open-Ended Professional Reports

Evaluating open-ended professional reports remains challenging due to the trade-off between stability and adaptivity.

LLM-as-a-judge methods (Zheng et al., 2023; Fan et al., 2025) enable flexible evaluation but suffer from stochastic variance and systemic biases (Li et al., 2024; Wang et al., 2024b). In contrast, reference-based and ground-truth matching approaches offer high verifiability but often fail to capture reasoning depth and evidential rigor.

Distinct from the above evaluation-time methods, a complementary line of research shapes model behavior at training time through human or AI feedback, including RLHF (Ouyang et al., 2022), RLAIF (Lee et al., 2024), and Constitutional AI (Bai et al., 2022). JADE operates in the inference-time evaluation setting, producing auditable assessments of completed reports grounded in expert skills, evidence verification, and explicit fact-to-reasoning dependencies. As an LLM-based judge, it also inherits known biases such as length and preference effects (Li et al., 2024; Dubois et al., 2025); we mitigate these via density normalization, deterministic skill activation, and structured, criterion-level scoring.

Recent work has explored checklist-based evaluation. Expert-authored rubrics (Arora et al., 2025; Ruan et al., 2026; Xu et al., 2025) provide rigorous standards but incur substantial human cost. Automated approaches such as DeepResearch-Bench (Du et al., 2026) rely heavily on LLM-generated checklists, weights and references, raising concerns about reliability, interpretability, and expert calibration.

In contrast, JADE grounds evaluation in deterministic, expert-designed skills and performs dynamic, claim-level verification. By decoupling domain-general principles from case-specific evidence, JADE achieves stable, interpretable, and adaptive assessment of professional reports.

# 3. JADE: Judge Agents with Dynamic Evaluation

## 3.1. Notation and Overview

Human experts do not evaluate reports by matching them against predefined ideal answers. Instead, they apply domain-specific principles and dynamically assess the validity of claims. JADE mirrors this cognitive process through a two-layer decomposition, enabling stable yet adaptive evaluation with minimal expert effort.

Formally, for each task we consider: a user query $q \in \mathcal{Q}$; an agent response $r \in \mathcal{R}$; a set of expert-authored evaluation skills $\mathcal{S} = \{s_1, \ldots, s_K\}$; a query-specific checklist $\mathcal{L}_q$ derived from $q$; and a report-specific checklist $\mathcal{L}_r$ derived from $(q, r)$, whose items are typed as either *evidence* (verifiable factual claims) or *reasoning* (judgment quality).

Each checklist $\mathcal{L} \in \{\mathcal{L}_q, \mathcal{L}_r\}$ consists of atomic Yes/No questions $\ell_i = (d_i, w_i)$, where $d_i$ is the question description and $w_i \in \mathbb{R}$ is the weight. Positive weights ($w_i > 0$) indicate quality requirements, while negative weights ($w_i < 0$) indicate critical flaws that penalize the score when triggered.

Given $(q, r)$, JADE produces a final score $S(q, r)$ and structured feedback through a fixed evaluation pipeline (Figure 2).

## 3.2. Layer 1: Query-Specific Checklist Generation

Each evaluation skill $s \in \mathcal{S}$ encodes a high-level professional principle (e.g., regulatory compliance). Each skill is associated with a rubric template $\mathcal{R}_s$ containing expert-authored checkpoints.

For a given query $q$, JADE activates a subset of relevant skills using a deterministic taxonomy-based mapping. In practice, this activation is implemented via multi-label classification, where skills are triggered either through expert pre-annotation or online prediction using learned binary or multi-class classifiers. We denote the activated skill set by:

$$\mathcal{S}_q \subseteq \mathcal{S}. \tag{1}$$

The query-level rubric is constructed by composing all activated templates:

$$\mathcal{R}(q) = \bigcup_{s \in \mathcal{S}_q} \mathcal{R}_s. \tag{2}$$

Given the composed rubric $\mathcal{R}(q)$, an LLM generates the **query-specific checklist**:

$$\mathcal{L}_q = \text{LLM}_{\text{gen}}^q(q, \mathcal{R}(q)) = \{\ell_1, \ldots, \ell_M\}. \tag{3}$$

Each checklist item $\ell_i = (d_i, w_i)$ consists of:

- $d_i$: an atomic Yes/No question (e.g., "Does the response provide product links?"),
- $w_i \in \mathbb{R}$: a weight where $w_i > 0$ for quality requirements and $w_i < 0$ for critical flaws,

Because skill activation depends only on $q$, the same query always activates the same skills and rubric, stabilizing checklist generation.

## 3.3. Layer 2: Report-Specific Checklist Generation

While $\mathcal{L}_q$ encodes query-level expectations, $\mathcal{L}_r$ captures report-specific claims and risks. Given a response $r$, an LLM generates the **report-specific checklist**:

$$\mathcal{L}_r = \text{LLM}_{\text{gen}}^r(q, r) = \{\ell_1^r, \ldots, \ell_P^r\}. \tag{4}$$

Each item $\ell_i^r$ is typed as either *evidence* or *reasoning*, yielding a typed decomposition:

$$\mathcal{L}_r = \mathcal{L}_r^{\text{ev}} \cup \mathcal{L}_r^{\text{re}}. \tag{5}$$

**Evidence** items in $\mathcal{L}_r^{\text{ev}}$ phrase verifiable factual claims as Yes/No questions (e.g., entity existence, quantitative attributes, source attributions); **reasoning** items in $\mathcal{L}_r^{\text{re}}$ ask whether conclusions are supported, assumptions stated, or logical steps valid. Evidence items form the factual base verified by the Verification Agent, while reasoning items are evaluated by the LLM Judge under evidence-aware gating.

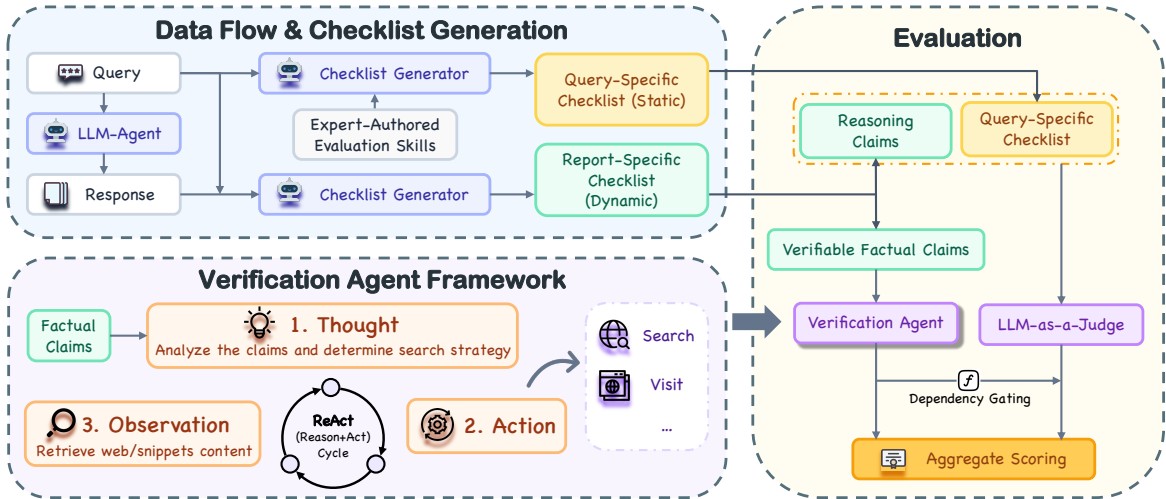

*Figure 2.* **Overview of JADE.** Given a query and an agent-generated response, JADE first activates appropriate expert-authored skills to guide the generation of query-specific checklists. It then derives report-specific checklists for verifiable factual claims and reasoning quality. Factual claims are validated via real-time web verification, while reasoning is assessed by LLMs conditioned on the query-specific checklists, with evidence-based gating to ensure that unsupported facts invalidate dependent judgments.

### 3.4. Scoring: Verification Agent and LLM Judge

JADE employs a two-track scoring mechanism:

**Evidence Verification.** For each evidence item $\ell \in \mathcal{L}_r^{\text{ev}}$, a **Verification Agent** performs real-time web search and content analysis to assign a verification score:

$$V(\ell) = \text{Agent}_{\text{verify}}(\ell) \in [0, 1]. \qquad (6)$$

**Reasoning Judgment.** For each query-level checklist item and reasoning-typed report item, $\ell_i \in \mathcal{L}_q \cup \mathcal{L}_r^{\text{re}}$, an **LLM Judge** evaluates the response against the criterion:

$$s_i = \text{LLM}_{\text{judge}}(q, r, d_i) \in \{0, 0.5, 1\}, \qquad (7)$$

where $d_i$ is taken from either $\mathcal{L}_q$ or $\mathcal{L}_r^{\text{re}}$ and 1 indicates "Yes", 0.5 indicates "Partial", and 0 indicates "No". For reasoning checklist items that depend on factual claims, the LLM Judge is additionally conditioned on the verification results to ensure evidence-aware assessment.

### 3.5. Dependency Gating

Each judged checklist item $\ell_i \in \mathcal{L}_q \cup \mathcal{L}_r^{\text{re}}$ may depend on a subset of evidence items:

$$\mathcal{D}(\ell_i) \subseteq \mathcal{L}_r^{\text{ev}}. \qquad (8)$$

JADE applies evidence-aware gating to ensure that conclusions built on unverified facts do not contribute to the score:

$$s_i^{\text{gated}} = \begin{cases} 0, & \exists \ell' \in \mathcal{D}(\ell_i) \text{ with } V(\ell') < \tau, \\ s_i, & \text{otherwise,} \end{cases} \qquad (9)$$

where $\tau$ is the verification confidence threshold. This mechanism prevents conclusions supported by invalid evidence from contributing to the final score.

### 3.6. Final Scoring and Theoretical Properties

JADE separately aggregates reasoning quality and evidence reliability.

**Reasoning Score.** We aggregate signed contributions, rewarding satisfied requirements and penalizing triggered critical flaws:

$$S_{\text{reason}}(q, r) = \frac{\sum_{\ell_i \in \mathcal{L}_q \cup \mathcal{L}_r^{\text{re}}} w_i \, s_i^{\text{gated}}}{\sum_{\ell_i \in \mathcal{L}_q \cup \mathcal{L}_r^{\text{re}}} w_i^+} \in [-\alpha, 1], \qquad (10)$$

where $w_i^+ = \max(w_i, 0)$ and $\alpha = \sum_i |w_i^-| / \sum_i w_i^+$ with $w_i^- = \min(w_i, 0)$. A perfect response (positives satisfied, no flaw triggered) reaches 1; each triggered flaw $w_i < 0$ subtracts $|w_i| / \sum_j w_j^+$.

**Evidence Score.** The evidence score is the mean verification reliability, with $V(\ell)$ incorporating the verifier's verdict and confidence:

$$S_{\text{evid}}(r) = \frac{1}{|\mathcal{L}_r^{\text{ev}}|} \sum_{\ell \in \mathcal{L}_r^{\text{ev}}} V(\ell) \in [0, 1]. \qquad (11)$$

**Overall Score.** The final evaluation score is defined as

$$S(q, r) = \max\big(S_{\text{reason}}(q, r), 0\big) \cdot S_{\text{evid}}(r) \in [0, 1], \qquad (12)$$

where clipping prevents negative reasoning from being inverted by reliable evidence, while the unclipped $S_{\text{reason}}$ is retained for ranking and diagnostics. This multiplicative

form ensures that strong reasoning unsupported by reliable evidence, or reliable evidence paired with poor reasoning, is consistently penalized.

JADE satisfies three key properties: (a) **Stability**. The query-specific checklist $\mathcal{L}_q$ depends only on the query $q$; (b) **Adaptivity**. The report-specific checklist $\mathcal{L}_r$ varies with response content; (c) **Soundness**. Checklist items depending on invalid evidence receive zero score via gating. The central design principle is therefore compositional: stable expert-grounded criteria and adaptive report-grounded verification are handled by separate layers, while dependency gating connects them through explicit fact-to-reasoning constraints.

# 4. BizBench: A Benchmark for Validating JADE

To empirically validate JADE under realistic professional workflows, we construct **BizBench**, a benchmark of open-ended business queries that require analytical reporting, evidence-grounded reasoning, and decision-making under real-world constraints. BizBench operationalizes JADE's framework through expert-authored skills, rubrics, and verification pipelines on authentic domain data.

Unlike factual QA benchmarks with fixed reference answers, BizBench targets settings where multiple structurally distinct reports may be valid and where external conditions evolve over time, rendering static ground-truth-based evaluation ill-posed.

## 4.1. Domain and Data Collection

BizBench focuses on *strategic sourcing*, characterized by heterogeneous information requirements, strong temporal dynamics, and open-ended professional reporting. We start from approximately 10,000 naturally occurring B2B queries (Figure 3). Automated deduplication, de-identification, and noise filtering reduce this pool to roughly 3,200 candidates. Domain experts then screen for analytical depth and multi-step reasoning, yielding about 350 candidate tasks, followed by final validation for professional realism and challenge level. This process yields 150 high-quality queries reflecting authentic expert workflows.

## 4.2. Hierarchical Task Taxonomy

Each query $q$ is annotated with a hierarchical multi-label taxonomy capturing professional task structure:

$$\mathcal{T}(q) = \mathcal{T}_1(q) \cup \mathcal{T}_2(q) \cup \mathcal{T}_3(q), \qquad (13)$$

where $\mathcal{T}_1(q)$ denotes primary intent, $\mathcal{T}_2(q)$ specifies information needs, and $\mathcal{T}_3(q)$ encodes operational constraints. Multiple labels may be assigned at each level.

*Table 1.* Summary statistics of BizBench.

| Statistic | Value |
|---|---|
| # Queries | 150 |
| Avg. query length (tokens) | 113.2 |
| Min/Max query length | 16 / 934 |
| Avg. # labels per query | 5.0 |
| # Taxonomy categories ($\mathcal{T}_1/\mathcal{T}_2/\mathcal{T}_3$) | 4 / 7 / 6 |
| Total Languages | 5 |

## 4.3. Instantiation of JADE Skills

The taxonomy supports JADE's taxonomy-guided skill activation mechanism. Each label $\lambda \in \mathcal{T}(q)$ corresponds to an evaluation skill $s_\lambda \in \mathcal{S}$. We define a mapping

$$\Gamma : 2^{\mathcal{T}} \to 2^{\mathcal{S}}, \qquad (14)$$

yielding the activated skill set

$$\mathcal{S}_q = \Gamma(\mathcal{T}(q)) \subseteq \mathcal{S}. \qquad (15)$$

The activated skills are composed into a query-level rubric:

$$\mathcal{R}(q) = \bigcup_{s \in \mathcal{S}_q} \mathcal{R}_s, \qquad (16)$$

which guides generation of the query-specific checklist $\mathcal{L}_q$.

## 4.4. Benchmark Characteristics

Table 1 summarizes key statistics of BizBench. Detailed distributions are provided in Appendix A. BizBench poses four systematic challenges: (a) **Real-world authenticity**: queries drawn from genuine B2B workflows preclude clean reference answers; (b) **Open-endedness**: multiple valid reports per query; (c) **Hierarchical complexity**: concurrent objectives and constraints; (d) **Temporal sensitivity**: time-dependent evidence verification. Appendix J details the comparison with existing benchmarks.

These properties motivate JADE's two-layer design: Layer 1 encodes stable, reusable skills, while Layer 2 adapts criteria to report-specific claims and evidence.

# 5. Experiments

## 5.1. Setup

**Evaluated Agents.** We evaluate representative agent systems spanning frontier APIs (GPT, Claude, DeepSeek, Gemini, Qwen) and commercial products (ChatGPT, Gemini). Here, "Shopping Research" denotes ChatGPT's shopping research mode, evaluated separately from ChatGPT with general web search. All systems were evaluated in January 2026; since closed-source backends may evolve over time, we cite official system cards, release notes, technical reports, or product documentation where available (OpenAI,

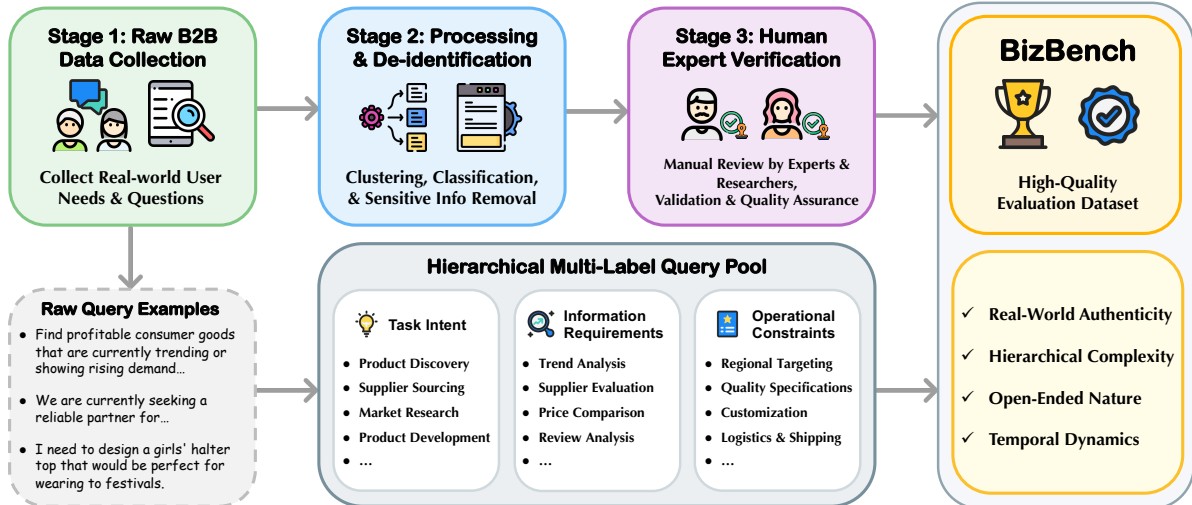

*Figure 3.* **Overview of BizBench.** Queries are collected from real B2B sourcing and market research scenarios, filtered and de-identified, and verified by domain experts. Each query is annotated using a hierarchical taxonomy that instantiates JADE's dynamic evaluation skills.

2024; 2025a;b;c;d; Google, 2025a;b; Anthropic, 2025a;b; DeepSeek-AI, 2025; Qwen Team, 2025).

**JADE Configuration.** JADE uses GPT-5-0807 as the backbone for checklist generation and judgment. Evidence verification employs a GenAI-tool-based verifier using Google Search and URL context extraction. We run each evaluation three times and report mean scores, with checklist weights set to $\{5, 10, -15\}$ by default.

**Metrics.** Beyond the Reasoning Score $S_{\text{reason}}$, Evidence Score $S_{\text{evid}}$, and final score $S$ defined in Section 3, we report a knowledge density indicator

$$U(q, r) = \frac{S(q, r)}{\log(\text{Tokens}(r) + 1)}, \qquad (17)$$

which normalizes $S$ by response length to reduce verbosity bias, and a source credibility metric based on source-website authority (Appendix G).

**Expert Effort.** Unlike per-query rubric authoring used by HealthBench (Arora et al., 2025) (48,562 expert criteria from 262 physicians over 11 months) or DR.BENCH (Yao et al., 2025b) (per-query rubrics refined through three rounds of expert review), JADE encodes evaluation principles once per domain. For BizBench, this amounted to less than 15 person-days for a 17-skill taxonomy reused across all 150 queries; Appendix B reports the full comparison.

## 5.2. Overall Performance

Table 2 presents the BizBench leaderboard. The average final score is **42.6%**, indicating substantial room for improvement toward expert-level quality. The top performers – Gemini Deep Research (57.1%) and Shopping Research (56.2%)—still fall short of the 80%+ threshold expected for

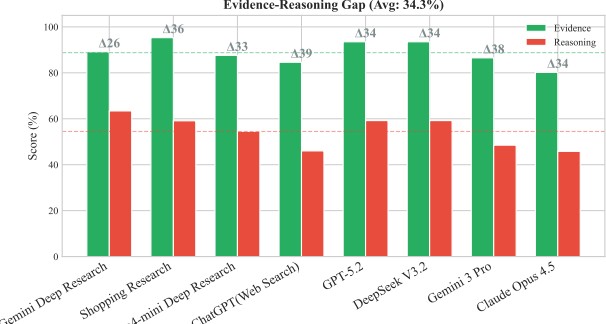

*Figure 4.* Evidence–Reasoning Gap for eight representative configurations; the in-figure average is computed over the displayed configurations only.

professional deliverables. This highlights a substantial gap between current agent capabilities and professional deployment requirements.

The leaderboard reveals clear stratification: agentic deep research systems outperform API-based agents, with Gemini Deep Research leading at 57.1%. Open-source models (DeepSeek V3.2) match proprietary alternatives.

## 5.3. Key Failure Modes

JADE's structured evaluation reveals two systematic failure modes invisible to holistic evaluation.

**Finding 1: Evidence-Reasoning Gap.** Across all 18 configurations in Table 2, we observe a striking disparity between evidence retrieval and professional reasoning (Figure 4). The average evidence score is **84.0%**, while reasoning averages only **50.5%**—a gap of 33.5 percentage points. This gap manifests concretely: for a market trend analysis query,

*Table 2.* BizBench Leaderboard. All experiments were performed in triplicate, and the mean was reported.

| Model | Type | Tool | Final (%) | Reasoning (%) | Evidence (%) | Credibility (%) | Density | Tokens |
|---|---|---|---|---|---|---|---|---|
| *Agentic Deep Research Systems* | | | | | | | | |
| Gemini Deep Research | Prop. | ✓ | **57.1** | **63.4** | 89.1 | 43.8 | 0.063 | 7590 |
| Shopping Research | Prop. | ✓ | 56.2 | 59.1 | **95.3** | 44.4 | 0.067 | 4772 |
| o4-mini Deep Research | Prop. | ✓ | 47.7 | 54.5 | 87.6 | 41.2 | 0.061 | 2907 |
| ChatGPT(Web Search) | Prop. | ✓ | 38.8 | 46.0 | 84.6 | 48.6 | 0.051 | 2244 |
| *API-Based Models with Tool Use* | | | | | | | | |
| GPT-5.2 | Prop. | ✓ | 55.7 | 59.2 | 93.5 | **50.2** | **0.071** | 3167 |
| DeepSeek V3.2 | Open | ✓ | 55.8 | 59.2 | 93.5 | 48.1 | 0.071 | 2877 |
| Gemini 3 Pro | Prop. | ✓ | 41.8 | 48.5 | 86.5 | 44.7 | 0.059 | 1419 |
| Claude Opus 4.5 | Prop. | ✓ | 36.2 | 45.8 | 80.2 | 44.8 | 0.049 | 1824 |
| Claude Sonnet 4.5 | Prop. | ✓ | 32.9 | 45.5 | 73.3 | 48.1 | 0.046 | 1975 |
| Qwen3-Max | Open | ✓ | 34.0 | 39.9 | 83.8 | 45.5 | 0.047 | 1340 |
| GPT-4.1 | Prop. | ✓ | 32.7 | 40.2 | 81.4 | 45.7 | 0.046 | 1434 |
| *API-Based Models (No Tool)* | | | | | | | | |
| GPT-5.2 | Prop. | – | 49.0 | 52.6 | 93.1 | **53.7** | 0.064 | 2516 |
| DeepSeek V3.2 | Open | – | 46.1 | 50.7 | 90.9 | 52.1 | 0.061 | 2299 |
| Gemini 3 Pro | Prop. | – | 44.8 | 52.7 | 84.8 | 50.7 | 0.063 | 1495 |
| GPT-4.1 | Prop. | – | 42.7 | 54.6 | 80.0 | 51.2 | 0.060 | 1380 |
| Qwen3-Max | Open | – | 34.2 | 46.0 | 74.0 | 50.9 | 0.051 | 1040 |
| Claude Opus 4.5 | Prop. | – | 32.1 | 49.3 | 68.0 | 52.2 | 0.044 | 2314 |
| Claude Sonnet 4.5 | Prop. | – | 29.1 | 41.8 | 71.6 | 52.2 | 0.041 | 1585 |

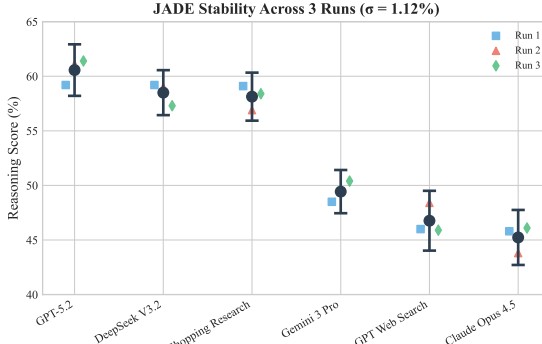

*Figure 5.* JADE stability across 3 runs. Error bars show $\pm 2\sigma$.

*Table 3.* Human alignment and ablation.

| Configuration | Human $r$ | Human $\rho$ | $\Delta$ vs Base | Variance |
|---|---|---|---|---|
| Vanilla | 0.667 | 0.736 | — | 1.45% |
| + Checklist | 0.769 | 0.758 | +15.3% | 1.45% |
| + Skill Only | 0.844 | 0.800 | +26.5% | 1.35% |
| + Report Only | 0.786 | 0.797 | +17.8% | 1.28% |
| JADE | **0.858** | **0.862** | **+28.6%** | **1.12%** |
| JADE[†] | 0.827 | 0.826 | +24.0% | 1.18% |
| JADE[‡] | 0.841 | 0.833 | +26.1% | 1.20% |

Vanilla: LLM-as-a-Judge, Pointwise
[†]: JADE with GPT-4.1 backbone.
[‡]: JADE with Gemini-3-flash backbone.

some systems are able to retrieve highly accurate data (evidence: 100%) but provide limited synthesis (reasoning: 5.3%). This indicates that current agents excel at information gathering but struggle with professional synthesis.

**Finding 2: Source Credibility Deficiency.** While agents retrieve factual content accurately (average evidence score: 84.0%), source credibility remains substantially lower (48.2%), revealing a **35.8% gap**. This reflects systematic difficulty in selecting authoritative and query-appropriate sources. In JADE, credibility evaluates whether citations originate from domain-authoritative primary sources (e.g., official filings, government statistics, original reports) rather than secondary or unverifiable webpages. We find that many agents rely heavily on low-credibility intermediaries instead of directly analyzing original documents, resulting in weakened evidential grounding. This *citation laundering* phe-

nomenon undermines professional trustworthiness despite factual correctness. Detailed scoring protocols are provided in Appendix G.

### 5.4. Stability: Expert-Aligned Rigorous Evaluation

We define **stability** as rigorous alignment with expert professional standards—not merely low variance across runs, but convergence toward calibrated, discriminative judgments that match human expert assessments. This section validates that JADE achieves such rigorous evaluation.

**Expert Alignment.** We conduct a controlled human alignment study on 180 reports (30 tasks × 6 models), engaging 5 domain experts for detailed annotation (IAA uses all experts without trimming; Appendix I). Full JADE achieves $r = 0.858$ Pearson correlation with human experts (Table 3), a **28.6% improvement** over vanilla LLM-as-Judge ($r = 0.667$). This strong alignment demonstrates that

JADE's evaluations converge toward professional standards rather than arbitrary LLM preferences. Refer to Appendix I for detailed experimental configurations.

**Model Robustness.** JADE remains effective across different backbones, achieving $r = 0.827$ (GPT-4.1) and $r = 0.841$ (Gemini-3-flash). Both variants outperform the vanilla baseline and stay close to the main setting, indicating strong model-agnostic robustness.

**Score Calibration.** Beyond correlation, rigorous evaluation should avoid systematically inflated absolute scores. Vanilla LLM-as-Judge tends to score substantially higher than human experts on average, while Full JADE produces stricter scores that better reflect professional quality levels. Similar trends are observed for GPT-4.1 and Gemini-3-flash, suggesting that this behavior is not tied to a single backbone.

**Reproducibility.** Low variance across runs ($\sigma = 1.12\%$; Figure 5) emerges as a byproduct of principled, expert-grounded evaluation. Deterministic skill activation eliminates arbitrary variation, while structured decomposition bounds the stochasticity inherent in LLM-based judgment.

**Component Contributions.** Ablation analysis reveals how each layer contributes to rigorous evaluation:

(a) **Skill Activation (Layer 1)**: Improves alignment from $r = 0.769$ to $r = 0.844$ (**+9.8%**). Expert-derived skills encode profession-specific requirements—compliance verification, quantitative metrics, operational specifics—that distinguish rigorous assessment from surface judgment.

(b) **Report-Specific Checklist (Layer 2)**: Improves alignment from $r = 0.769$ to $r = 0.786$ (**+2.2%**). This component enables claim-level verification that catches citation hallucination and unsupported assertions—errors that holistic evaluation overlooks.

(c) **Synergy**: Combining both layers yields an additional +1.4% improvement, confirming that the two-layer architecture achieves more than the sum of its parts.

The **+ Checklist** baseline corresponds to a flat rubric setting with generated criteria but without JADE's structured decomposition. Its lower alignment ($r = 0.769$) shows that the gain does not come simply from using a longer checklist. Skill activation narrows evaluation to professionally relevant criteria, while dependency gating imposes explicit evidence–reasoning constraints that a flat independent rubric cannot express.

**Operational Cost.** Complementing the one-time domain effort discussed in the Setup above, JADE incurs higher per-query inference cost than a single-call judge because it explicitly decomposes checklist generation, scoring, verification, and gating. Table 4 reports the operational cost under our recommended setup: GPT-5-0807 for checklist

*Table 4.* Operational cost per query–response evaluation.

| Evaluator | Calls | Cost | Human $r$ |
|---|---|---|---|
| Vanilla judge | 1.0 | $0.004–$0.013 | 0.667 |
| JADE w/o verification | 15.6 | $0.057 | 0.844 |
| **JADE** | **19.6** | **$0.126** | **0.858** |
| DR.BENCH protocol | ∼76 | $0.108 | – |

*Table 5.* DR.BENCH cross-domain validation. Selected domains are shown; Appendix C reports all 10 domains. Sports & Competitions is the only non-significant domain ($p = 0.297$).

| Domain / group | n | Pearson $r$ | Spearman $\rho$ |
|---|---|---|---|
| All domains | 214 | 0.736 | 0.789 |
| Environment & Sustainability | 12 | 0.882 | 0.941 |
| Business & Finance | 35 | 0.733 | 0.848 |
| Health & Medicine | 19 | 0.758 | 0.609 |
| Sports & competitions | 15 | 0.478 | 0.300 |

generation and Gemini-3-flash for scoring and verification. Each query–response pair costs about $0.126 across 19.6 LLM calls on average, compared with $0.004–$0.013 for a vanilla pointwise judge. This cost buys substantially higher human alignment ($r = 0.858$ vs. 0.667) and more calibrated scores, as reported above. Since only activated skills are evaluated (5.0 labels per query on average, out of 17 total skills; Appendix B), cost scales with task complexity rather than the full domain taxonomy.

## 5.5. Adaptivity: Cross-Domain and Report-Specific Evaluation

**Adaptivity** in JADE manifests in two dimensions: (i) *cross-domain generalization*—the ability to transfer evaluation principles to new professional domains, and (ii) *report-specific evaluation*—dynamic adaptation of criteria based on response content. We validate both dimensions below.

**Cross-Domain Transfer.** We first evaluate JADE on DR.BENCH (Yao et al., 2025b), which contains 214 expert-curated deep-research tasks across 10 domains. To test whether JADE's compositional principle transfers beyond BizBench, we use domain-level skills auto-summarized by GPT-5-0807 from DR.BENCH's rubrics, with no per-query rubrics and no expert-authored skill curation. This deliberately conservative setup provides a lower-bound transfer test, while the stronger BizBench alignment with expert-authored skills (Table 3) suggests that human skill curation remains valuable.

As shown in Table 5, JADE achieves strong overall alignment with DR.BENCH's expert rubric scores (Pearson $r = 0.736$, Spearman $\rho = 0.789$), with statistically significant correlations in 9 of 10 domains. The strongest domains are open-ended analytical settings such as environment, history, law, business, and education; the weakest domain is sports, where many rubric items require recalling

*Table 6.* HealthBench transfer validation. Spearman $\rho$ between JADE-generated and expert-authored checklist scores. *With Skill* uses theme-summarized domain skills; *No Skill* omits domain adaptation. Trim removes outlier checklist items containing highly specific clinical details (e.g., exact dosages).

| Dataset | With Skill | No Skill | Improvement |
|---|---|---|---|
| Full (n=474) | 0.228 | 0.154 | +48% |
| Trim 10% (n=422) | 0.379 | 0.083 | +358% |
| Trim 20% (n=371) | 0.530 | 0.187 | +183% |

a fixed set of named entities. These results support JADE's cross-domain generality while also identifying a principled boundary: domain skills encode professional evaluation principles, not exhaustive closed-form answer keys.

As a complementary transfer test in a high-stakes medical domain, we further evaluate JADE on HealthBench (Arora et al., 2025), a benchmark with 1000 hard-level tasks with expert-authored checklists across 7 clinical themes (emergency referrals, global health, clinical communication, etc.). We select a subset of 474 instances whose checklists contain at least five items for evaluation.

HealthBench provides ground-truth checklists written by medical professionals for each query. To instantiate JADE in medicine, we use GPT-5-0807 to auto-summarize theme-level principles from these expert-authored checklists and use them as domain skills—mirroring the conservative skill-construction recipe used for DR.BENCH, so that no per-query expert rubrics enter JADE. We then compare scores from JADE-generated checklists with scores from expert-authored checklists on the 474-instance subset, and report their Spearman correlation ($\rho$).

As shown in Table 6, on the full 474 tasks, adding skills improves $\rho$ from 0.154 to 0.228 (+48%). After proportionally removing 10% outliers within each clinical theme—items containing highly domain-specific clinical details like exact dosages ("1 mg IV/IO every 3-5 minutes") or guideline classifications ("Class I, Level A") that require specialized medical training—alignment improves substantially to $\rho = 0.379$ (+358%). With 20% trimming, performance reaches $\rho = 0.530$ (+183%). These results demonstrate the potential of theme-specific skill optimization and validate that JADE's *principle* of encoding expert knowledge as reusable evaluation dimensions generalizes across professional domains. More details are provided in Appendix D.

**Report-Specific Dynamic Evaluation.** Unlike static checklists that apply identical criteria regardless of response content, JADE generates report-conditioned evaluation criteria that adapt to each response's specific claims, evidence patterns, and reasoning structure. This enables detection of response-specific failures invisible to fixed rubrics.

**Case Example: Evidence-Gated Reasoning.** Consider

an agent report claiming that a price range of $999–$1,199 CAD corresponds to approximately $730–$875 USD. JADE verifies this numerical premise using real-time exchange rates and detects a material deviation. As a result, the dependent pricing justification is gated and assigned zero credit, preventing inaccurate financial assumptions from influencing downstream reasoning. Additional details are provided in Appendix H.

### 5.6. Summary

Our experiments validate JADE's resolution of the stability–adaptivity dilemma. JADE achieves strong stability through expert-aligned, well-calibrated, and consistent evaluation. It demonstrates adaptivity via cross-domain transfer on DR.BENCH and HealthBench, as well as report-specific dynamic assessment. JADE further exposes systematic failure modes, including evidence–reasoning and source credibility gaps hidden by holistic evaluation.

## 6. Limitations

For deployment in a new domain, experts should first define a compact reusable skill taxonomy and verify that target tasks require open-ended professional judgment rather than exhaustive factual recall. This domain adaptation cost is nontrivial, and JADE's effectiveness may decrease in knowledge-intensive fields such as medicine and education. More generally, JADE is designed for open-ended professional judgment, where multiple valid response strategies can be evaluated against reusable principles. It is less suitable for closed-form factual recall tasks that require exhaustive answer keys, as reflected by weaker DR.BENCH results in sports-style named-entity recall. In addition, evaluation quality depends on rubric design and judge model reliability, and the multi-stage pipeline increases system complexity and cost. Although BizBench is highly challenging, its primarily single-turn and underspecified queries introduce sensitivity to response style, as some agents favor detailed one-shot reports while others rely on multi-turn interaction.

## 7. Conclusion

We propose **JADE**, a two-layer framework that resolves the stability–adaptivity dilemma in evaluating open-ended professional LLM outputs. Through expert-grounded skill activation and report-specific claim-level verification, JADE achieves both strong stability and adaptivity on BizBench, enabling cross-domain transfer and dynamic detection of response-specific failures. JADE also reveals evidence–reasoning gaps and citation hallucination missed by holistic evaluators.

## Impact Statement

JADE provides a more rigorous and transparent evaluation layer for open-ended professional reports produced by agentic AI systems. Its primary positive impact is to make agent evaluation more diagnostic: by decomposing judgments into expert-grounded criteria, claim-level verification, source credibility, and evidence-dependency gating, JADE exposes failure modes such as unsupported reasoning, hallucinated citations, weak source use, and citation laundering that holistic scores can miss. This can support safer research and deployment decisions in professional workflows where users need to understand not only whether an agent answer seems plausible, but which claims and reasoning steps are trustworthy. The public release of BizBench, rubrics, and evaluation code is intended to enable reproducible comparison and independent scrutiny of agent capabilities on temporally dynamic, reference-free tasks. At the same time, JADE should not be treated as a substitute for domain experts in settings where factual recall, legal obligations, medical advice, or safety-critical decisions require specialized professional review. Automated evaluation scores should not be misinterpreted as quality certifications: high JADE scores do not absolve practitioners from verifying critical claims, particularly in commercially consequential settings such as strategic sourcing, where evaluation outcomes may influence vendor rankings and procurement decisions. Its outputs depend on the quality of the expert skills, verifier, and external evidence sources used during evaluation. To support evaluation integrity, BizBench queries reflect naturally occurring professional inquiries and, to our knowledge, were not used in training any evaluated model.

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

# A. BizBench Dataset Details

## A.1. Multi-Label Taxonomy

BizBench uses a three-level hierarchical taxonomy to capture professional task structure. Each query is annotated with labels from multiple levels, reflecting the inherently multi-objective nature of B2B sourcing tasks.

$\mathcal{T}_1$ — **Primary Intent (4 categories).** The distribution of top-level intent is illustrated in Table 7, reflecting the user's primary goal.

*Table 7.* Primary intent ($\mathcal{T}_1$) distribution in BizBench.

| Category | Count | Description |
|---|---|---|
| Supplier Sourcing | 68 | Finding suppliers, manufacturers, or factories |
| Product Discovery | 42 | Finding products matching specifications |
| Market Research | 28 | Market trends, competitor analysis |
| Product Development | 12 | OEM/ODM, customization requirements |

$\mathcal{T}_2$ — **Information Need (7 categories).** The second level specifies the dimensions of required information:

- **Supplier Evaluation**: Credentials, certifications, capabilities
- **Price Comparison**: Price ranges, cost structures, margins
- **Review Analysis**: User reviews, ratings, feedback synthesis
- **Sales Data**: Sales volumes, rankings, market share
- **Trending Analysis**: Market trends, seasonal patterns
- **Platform Data**: E-commerce platform metrics, listings
- **Competitor Analysis**: Competitive landscape, positioning

$\mathcal{T}_3$ — **Operational Constraint (6 categories).** The third level reveals the relevant constraints.

- **MOQ/Price Constraint**: Minimum order quantities, budget limits
- **Certification Required**: CE, FDA, ISO, organic certifications
- **Region Specific**: Geographic targeting, local sourcing
- **Customization/OEM**: Custom branding, design modifications
- **Quality Specification**: Material, durability, technical specs
- **Logistics/Shipping**: Shipping terms, lead times, Incoterms

## A.2. Query Examples

Representative queries and their corresponding multi-labels are presented below. Here, JSON fields L1–L3 correspond to taxonomy levels $\mathcal{T}_1$–$\mathcal{T}_3$.

**Example Queries**

```
[
  {
    "id": 8,
    "query": "Source a coffee machine accessories, logo branding, and support
            for small-batch OEM. Prioritize suppliers with export experience,
            CE certifications. I want to sell these at Amazon US market.",
    "L1_primary_intent": "supplier_sourcing",
    "L2_information_need": [
```

```
        "supplier_evaluation",
        "platform_data"
      ],
      "L3_constraints": [
        "certification_required",
        "customization_oem",
        "region_specific"
      ]
    },
    {
      "id": 16,
      "query": "Give me a comprehensive list of at least 80 hot-selling and uniquely
                designed kitchenware products (such as cooking pots, pans, bowls, etc.)
                that are currently popular on Amazon.",
      "L1_primary_intent": "product_discovery",
      "L2_information_need": [
        "trending_analysis",
        "platform_data"
      ],
      "L3_constraints": []
    },
    {
      "id": 27,
      "query": "Analysis of market opportunities in the European vegan cheese sector,
                identifying consumer trends and potential growth areas",
      "L1_primary_intent": "market_research",
      "L2_information_need": [
        "trending_analysis"
      ],
      "L3_constraints": [
        "region_specific"
      ]
    },
    {
      "id": 43,
      "query": "Alright, what I want to create is the Cool Pets Australia Christmas
                Pet box. I wanna create an epic product here that can be created,
                produced, and packaged and sent to Australia immediately.\n\nWhat I
                wanna do is work with a company that can drop ship or just send a heap
                to Australia. Not necessarily Christmas themed, but we can try and
                market them for the Christmas rush. What we wanna do is just create an
                absolute viral frenzy with it. I wanna go at the higher price point,
                so I wanna include cheap things that I can throw in this box that are
                gonna be of interest to pet owners. And I wanna be able to do it so we
                can promote Cool Pets Australia as well.\n\nLike a box, it'll be a
                mystery box almost. We could price it at $350. It's got some good
                things in there that the average everyday pet owner needs for their
                cat or their dog, and we can tailor it. Small dog owner large dog
                owner cat owner",
      "L1_primary_intent": "product_development",
      "L2_information_need": [
        "price_comparison",
        "platform_data",
        "trending_analysis"
      ],
      "L3_constraints": [
        "moq_price_constraint",
        "logistics_shipping",
        "region_specific",
        "customization_oem"
      ]
    }
  ]
```

## B. Expert Effort and Deployment Cost

JADE requires experts to define reusable domain skills rather than per-query rubrics. For BizBench, this process produced 17 structured skill templates organized under a 4/7/6 three-level taxonomy. The estimated one-time effort was less than 15 person-days: roughly 2–3 person-days for taxonomy design and about half a day per skill template once the schema was fixed. The resulting skill activation is sparse: each query activates 5.0 labels on average out of 17 total skills. Table 8 compares this pattern with benchmarks that rely on per-query rubric construction.

*Table 8.* Expert effort comparison.

| Framework | Expert artifact | Granularity | Reported scale |
|---|---|---|---|
| HealthBench (Arora et al., 2025) | Expert-authored checklist items | Per query | 48,562 criteria |
| DR.BENCH (Yao et al., 2025b) | Query-specific and general report rubrics with expert review | Per query | Rubrics for 214 tasks |
| JADE / BizBench | Domain taxonomy and reusable skills | Per domain | 17 skills, <15 person-days |

This does not remove the need for domain expertise: deploying JADE in a new field still requires experts to decide which principles matter and where evaluator competence ends. However, once a domain skill library exists, the same skills can be reused across many queries and updated when domain standards change.

**Deployment guideline.**    For a new domain, practitioners should first identify whether the target tasks involve open-ended professional judgment with multiple valid solution paths. If the task instead requires exhaustive recall of a fixed answer set, an answer-key or retrieval-based evaluator is more appropriate. When JADE is suitable, experts should define a compact taxonomy of reusable skills, validate the skill activations on a small labeled set, and update skills as domain standards change.

## C. DR.BENCH Cross-Domain Validation

To evaluate cross-domain generality beyond strategic sourcing, we first conduct validation on DR.BENCH (Yao et al., 2025b). DR.BENCH contains 214 expert-curated deep-research tasks spanning 10 domains. For this experiment, JADE uses domain-level skills auto-summarized by GPT-5-0807 from DR.BENCH's rubrics, and matches DR.BENCH's uniform 0/1/2 scoring format. This setup is intentionally conservative: it uses no per-query rubrics and no expert-authored skills, so it estimates a lower bound for JADE's transfer performance.

*Table 9.* DR.BENCH domain-level alignment.

| Domain | n | Pearson $r$ | Spearman $\rho$ | $p(\rho)$ |
|---|---|---|---|---|
| Environment & Sustainability | 12 | 0.882 | 0.941 | < 0.001 |
| History & Social Sciences | 33 | 0.830 | 0.877 | < 0.001 |
| Law & Politics | 24 | 0.814 | 0.874 | < 0.001 |
| Business & Finance | 35 | 0.733 | 0.848 | < 0.001 |
| Commonsense & Education | 15 | 0.752 | 0.833 | < 0.001 |
| Academia & Research | 23 | 0.709 | 0.824 | < 0.001 |
| News & Current Affairs | 18 | 0.751 | 0.812 | < 0.001 |
| Technology Intelligence | 17 | 0.777 | 0.707 | 0.002 |
| Health & Medicine | 19 | 0.758 | 0.609 | 0.006 |
| Sports & Competitions | 15 | 0.478 | 0.300 | 0.297 |
| **All** | **214** | **0.736** | **0.789** | < 0.001 |

The results show strong alignment in domains dominated by open-ended professional judgment, including environment, law, history, business, education, academia, and news analysis. The weaker Sports & Competitions result reveals a clear boundary: many rubric items require exact recall of a fixed list of named entities, which is better handled by answer-key or fact-retrieval evaluation than by reusable domain principles. The Health & Medicine result is also lower than open-ended domains because some criteria require exact clinical dosages, guideline classifications, or procedural details.

**Lower bound vs. expert-authored skills.**    The DR.BENCH protocol here uses LLM-summarized skills as a deliberately conservative substitute for human curation, in contrast to BizBench's expert-authored skill library (Appendix B). Although

*Table 10.* HealthBench theme distribution.

| Theme | Count | Description |
|---|---|---|
| Global Health | 135 | Adapt responses to account for variations across multilingual environments |
| Context Seeking | 81 | Proactively solicit supplementary details when necessary |
| Hedging | 71 | Distinguish between reducible and irreducible uncertainties |
| Health Data Tasks | 66 | Safely execute specific health data tasks |
| Communication | 44 | Identify whether the user is a medical professional |
| Complex Responses | 43 | Provide responses with an appropriate level of granularity |
| Emergency Referrals | 34 | Evaluate whether the model can accurately triage |
| **Total** | **474** | |

*Table 11.* HealthBench trimming distribution.

| Theme | Full | Trim 10% | Trim 20% |
|---|---|---|---|
| Global Health | 135 | 121 | 106 |
| Context Seeking | 81 | 72 | 63 |
| Hedging | 71 | 63 | 56 |
| Health Data Tasks | 66 | 59 | 52 |
| Communication | 44 | 39 | 34 |
| Complex Responses | 43 | 38 | 34 |
| Emergency Referrals | 34 | 30 | 26 |
| **Total** | **474** | **422** | **371** |

the two results are not a controlled same-dataset ablation, they provide complementary evidence: JADE obtains strong alignment even with auto-summarized skills (Pearson $r = 0.736$, Spearman $\rho = 0.789$), while the higher BizBench alignment with expert-authored skills (Pearson $r = 0.858$, Spearman $\rho = 0.862$) is consistent with the value of expert curation for high-stakes deployment.

# D. HealthBench Transfer Experiments

We further examine transfer in a high-stakes medical setting using HealthBench (Arora et al., 2025).

## D.1. Dataset Overview

HealthBench consists of two subsets: Consensus and Hard. Focusing on the Hard subset, which initially contains 1,000 instances, we conducted a filtering process to exclude multi-turn dialogues and samples with five or fewer native rubrics. This refined collection comprises 474 medical queries across 7 clinical themes, with detailed statistics provided in Table 10.

## D.2. Theme-Level Correlation Analysis

Table 12 shows Spearman correlation with expert checklists broken down by clinical theme. Skill activation improves alignment in 5/7 themes. Emergency Referrals and Global Health show strongest improvement (+194% and +908% with trimming), reflecting well-defined professional protocols. Table 11 reports the proportional trimming distribution used in the main experiment. Within each theme, removed examples are those with the largest divergence between JADE-generated checklist scores and expert-authored checklist scores. Manual inspection shows that many removed items are dominated by exact clinical recall, such as precise dosages, guideline levels, or procedural codes, rather than reusable professional judgment principles.

**Theme-Level Insights:**

- **Emergency Referrals** shows highest correlation ($\rho = 0.438$ after trimming), as urgent care triage follows well-established protocols that skill-based evaluation captures effectively.

- **Health Data Tasks** shows improved correlation ($\rho = 0.225$ after trimming), though still lower than other themes, reflecting that clinical documentation requires specialized medical training.

Table 12. HealthBench theme-level Spearman correlation ($\rho$).

| Theme | n | Full Data | | Trim 10% | | Skill Wins(Full) |
|---|---|---|---|---|---|---|
| | | Skill | No Skill | Skill | No Skill | |
| Emergency Ref. | 34 | 0.287 | 0.194 | 0.438 | 0.149 | ✓ |
| Global Health | 135 | 0.240 | 0.128 | 0.393 | 0.039 | ✓ |
| Communication | 44 | 0.192 | 0.098 | 0.162 | -0.023 | ✓ |
| Hedging | 71 | 0.197 | 0.245 | 0.407 | 0.183 | ✗ |
| Context Seeking | 81 | 0.153 | 0.253 | 0.331 | 0.227 | ✗ |
| Complex Resp. | 43 | 0.099 | 0.014 | 0.160 | -0.012 | ✓ |
| Health Data | 66 | 0.102 | -0.010 | 0.225 | -0.048 | ✓ |
| **Overall** | **474** | **0.228** | **0.154** | **0.379** | **0.083** | ✓ |

- **Context Seeking** and **Hedging** themes show mixed results, as these involve subjective judgment about when to seek clarification or express uncertainty.

### D.3. Ablation: Skill vs. Evidence Components

To further verify the role of each JADE component on HealthBench, we use a subset of 175 samples (the first 25 samples per theme) and conduct a detailed ablation with evidence verification. Spearman correlations are calculated against scores derived from the HealthBench rubrics, as shown in Table 13. Removed samples exhibit the weakest correlation with the HealthBench rubric scores under the corresponding configuration, so we treat them as outliers for diagnostic analysis.

**Framework-Level Insights:**

- **Synergistic Efficacy of JADE:** The Full JADE configuration consistently outperforms all other variants, achieving a peak correlation of $\rho = 0.607$. This superiority underscores the necessity of both Evidence Verification and Skill-Based Reasoning; while evidence provides the essential factual grounding to prevent hallucinations, the skill module offers the structural logic required to interpret complex medical nuances. Removing either component leads to a significant performance degradation.

- **Sensitivity to Data Quality:** Correlation coefficients more than double (reaching $\rho = 0.607$) after removing a small number of outliers, suggesting that while JADE is highly effective for standard medical queries, extreme edge cases in HealthBench remain a challenge for current LLM-based evaluators.

Table 13. HealthBench ablation study.

| Configuration | Spearman $\rho$ | vs. Baseline |
|---|---|---|
| *Full dataset (n=175):* | | |
| Baseline (No Structure) | 0.075 | — |
| JADE w/o Skill | 0.061 | -19% |
| JADE w/o Evidence | 0.247 | +229% |
| **Full JADE** | **0.270** | **+260%** |
| *After remove 5 samples per theme (n=140):* | | |
| Baseline | 0.256 | — |
| JADE w/o Skill | 0.316 | +23% |
| JADE w/o Evidence | 0.551 | +115% |
| **Full JADE** | **0.560** | **+119%** |
| *After remove 7 samples per theme (n=126):* | | |
| Baseline | 0.314 | — |
| JADE w/o Skill | 0.412 | +31% |
| JADE w/o Evidence | 0.600 | +91% |
| **Full JADE** | **0.607** | **+93%** |

## E. Prompt Templates

This section presents the core prompt templates used in JADE's two-layer evaluation.

## E.1. Layer 1: Query Checklist Generation Prompt

The following prompt generates the query-specific checklist $\mathcal{L}_q = \text{LLM}_{\text{gen}}^q(q, \mathcal{R}(q))$, where $\mathcal{R}(q)$ is composed from activated expert-derived skills; parts of the prompt in the OUTPUT FORMAT have been omitted:

---

**Query Checklist Generation Prompt Template**

```
# TASK
Generate a checklist to evaluate if an AI response adequately answers the user
query.
Each criterion must be an atomic Yes/No question (ask ONE thing only).

# QUERY
{query}

# CORE DELIVERABLE (L1 Gate)
{deliverable_check}

# EXPERT CHECKPOINTS
{expert_hints}

# RULES
1. **L1 Gate First**: item_id=0 must check if core deliverable exists
    - Product/supplier queries  check for links (URLs, ASINs)
    - Data queries  check for specific data/numbers
    - Analysis queries  check for conclusions with reasoning

2. **Atomic Questions**: Each criterion = ONE check
    - BAD: "Does it provide links AND analyze trends?"
    - GOOD: "Does it provide product links?"
    - GOOD: "Does it analyze trends?"

3. **Quantity**: Scale with query complexity
    - Simple query (few requirements): 4-6 items
    - Complex query (many L2/L3 checkpoints): 8-15 items
    - Cover each expert checkpoint; Cover the user's requirements; Skip redundant or
      trivial checks

4. **Critical Flaw**: Only for ACTIVE violations (recommending wrong things)
    - GOOD: "Does it recommend items outside the specified scope?"
    - BAD: "Does it fail to provide X?" (covered by positive check)

5. **Independent Items** (depends_on: null): Always include these at the end
    - **Graceful Degradation**: If core request cannot be fully met, does the
      response acknowledge limitations and provide alternatives?
    - **Risk Awareness**: For recommendation/decision queries, does the response
      mention potential risks or uncertainties?

# OUTPUT FORMAT
Each item has: item_id, tier, depends_on, category, description, weight,
    source_skill

Weights: 15 (L1 core), 10 (L2/L3), 5 (general), -15 (critical flaw)

```json
[
  {{
     "item_id": 0, "tier": "L1", "depends_on": null, "category": "Core Deliverable",
     "description": "Does the response provide [SPECIFIC DELIVERABLE]?",
     "weight": 15, "source_skill": "L1"
  }},
  ...
]
```

---

## E.2. Layer 2: Report-Specific Checklist Generation Prompt

The following prompt generates the report-specific checklist $\mathcal{L}_r$, jointly producing evidence-typed items ($\mathcal{L}_r^{\text{ev}}$, verifiable factual claims phrased as Yes/No questions) and reasoning-typed items ($\mathcal{L}_r^{\text{re}}$); parts of the prompt in the OUTPUT FORMAT have been omitted:

---

**Report-Specific Checklist Generation Prompt Template**

```
# TASK
Generate a checklist to verify factual claims and reasoning quality in the AI
response.

# QUERY
{query}

# RESPONSE TO EVALUATE
{report_content}

# ITEM TYPES

1. **Evidence** (type: "evidence"): Verifiable facts
    - Factual claims (entity existence, specs, certifications)
    - Quantitative claims (numbers, dates, prices)
    - Source validity (URLs cited in response)

2. **Reasoning** (type: "reasoning"): Judgment quality
    - Is the conclusion supported by stated evidence?
    - Are key assumptions stated?
    - Is the reasoning logically valid?

# RULES
1. Each description must be SELF-CONTAINED (understandable without the response)
    - BAD: "Verify the Enaiter example's price claim"
    - GOOD: "Verify that Enaiter silicone products are priced at $7.50-$9.80/piece"

2. Focus on HIGH-IMPACT claims that affect user decisions

3. Quantity: 4-10 items based on response complexity

4. Reasoning items may include depends_on: evidence item_ids they rely on

# OUTPUT FORMAT
```json
[
  {{
    "item_id": 0, "type": "evidence", "category": "Factual Claim",
    "description": "Verify that [SPECIFIC CLAIM with full context].",
    "weight": 5
  }},
  {{
    "item_id": 2, "type": "reasoning", "category": "Evidence Support",
    "description": "Is [CONCLUSION] supported by the verified evidence?",
    "weight": 10, "depends_on": [0]
  }},
  ...
]
```
```

---

## E.3. Evidence Verification Prompt

The verification agent uses the following prompt to validate claims:

**Evidence Verification Prompt Template**

```
You are an expert fact-checker. Your task is to verify the following claim using
web search and URL context tools.

## Current Date: {current_date}

## Claim to Verify
{claim}

{source_info}

## Instructions
1. If a source URL is provided, analyze it first using URL context
2. Use web search to find additional evidence if needed
3. Analyze all gathered evidence carefully
4. Provide your final verdict in the EXACT JSON format below

## IMPORTANT: Handling Equivalent Terms and Partial Matches
When verifying claims, recognize that many terms have **equivalent expressions**.
Do NOT require the report/listing to repeat every alias verbatim if the meaning is
clearly the same.

1. **Parenthetical equivalence** like "18/8 (304)" or "X (Y equivalent)" means
   these are **equivalent terms**. Finding EITHER term satisfies the claim.
   - Example: "18/8 stainless steel" = "304 stainless steel" = "SUS304" (same
     material, different naming conventions)
   - If claim says "18/8 (304) stainless steel" and evidence shows only "18/8
     stainless steel", this is still a MATCH.

2. **Slash notations** like "USB-C/Type-C" are interchangeable terms.

3. **Common industry equivalents** you should recognize:
   - 18/8 stainless steel = 304 stainless steel = SUS304
   - WiFi 6 = 802.11ax
   - USB 3.0 = USB 3.1 Gen 1 = USB 3.2 Gen 1
   - 4K = 2160p = UHD

4. **Focus on semantic meaning**: if the evidence confirms the essential claim
   using an equivalent term, conclude "yes".

## Response Format (MUST be valid JSON)
```json
{{
    "conclusion": "yes" or "no",
    "confidence": 0-100,
    "reason": {{
        "summary": "Brief summary of your findings",
        "supporting": ["Evidence point 1", "Evidence point 2"],
        "contradicting": ["Contradicting evidence if any"]
    }},
    "reference_urls": {{
        "supporting": ["url1", "url2"],
        "contradicting": ["url3"]
    }}
}}
```

## Rules
- "yes" means the claim is accurate/verified
- "no" means the claim is inaccurate, outdated, or cannot be verified
- Confidence should reflect how certain you are (0-100)
- Include ALL relevant URLs you found in reference_urls
```

```
 - Be thorough but concise in your reasoning
```

## F. Skill Templates

Each label at all levels ($\mathcal{T}_1/\mathcal{T}_2/\mathcal{T}_3$) is associated with a corresponding skill in JADE's taxonomy. We list representative $\mathcal{T}_1$ (primary intent) skills with brief descriptions below. However, due to business reasons, we only provide a basic version of the skill in the code repository.

---

**$\mathcal{T}_1$ Skill Example: Supplier Sourcing**

```
name: "Supplier Sourcing"
description: "A comprehensive process for identifying, screening, and locking in
              suitable upstream suppliers, OEMs, or factories in global markets
              based on specific product requirements."

primary_deliverable:
  name: "Concrete Supplier Information"
  description: "A report containing verifiable data, not just entity names."
  must_have:
    - "Direct, verifiable URLs (Official Website, Alibaba/Global Sources Profile)"
    - "Direct contact details (WhatsApp, WeChat, direct email or phone number)"
    - "Key commercial terms: MOQ (Minimum Order Quantity), Lead Time, etc."

hints:
  - rule: "Must provide verifiable supplier links"

  - rule: "Distinguish factories from trading companies when possible"
    definitions:
      factory: "Owns production lines, lower price, higher MOQ."
      trading_company: "Middleman, higher price, lower MOQ, better service."

  - rule: "Include cost structure with clear assumptions"
    require:
      - "Estimated Domestic & International Freight"
      - "Import Duties (based on HS Code)"
      - "Landed Cost Formula: Unit Price + Freight + Duties + Insurance"

  - rule: "Mention supplier verification risks"
    examples:
      - "Payment Scams (Personal bank accounts)"
      - "Quality Fade (Golden sample vs. mass production)"
      - "Fake Certifications (Photoshop ISO/SGS docs)"
```

---

**$\mathcal{T}_2$ Skill Example: Price Comparison**

```
name: "Price Comparison"
description: "A systematic analysis of pricing hierarchies across the supply chain,
              aimed at establishing realistic cost baselines, identifying market
              positioning."

hints:
  - rule: "Specify price scope (unit/landed/retail vs wholesale)"
    reasoning: "Directly comparing a factory quote to a retail price is misleading
                without accounting for logistics and overhead."

  - rule: "Indicate currency and source of pricing"
    reasoning: "Exchange rates fluctuate, and platform listings (e.g., Alibaba vs.
                Amazon) have different validity periods."
    require:
      - "Standardize all pricing to a single reference currency."
```

```
        - "Cite the specific source."

   - rule: "Explain what drives price differences"
     examples:
        - "Material Grade (e.g., 304 Stainless Steel vs. 201 Stainless Steel)"
        - "MOQ Impact (Price at 100 units vs. Price at 10,000 units)"
        - "Compliance Costs (UL/CE certifications)"
        - "Packaging Quality (Brown box vs. 4-color gift box)"
```

### $\mathcal{T}_3$ Skill Example: Certification Required

```
name: "Certification Required"
description: "A critical compliance check to ensure products and manufacturers meet
             legal, safety and quality standards for specific target markets (e.g.,
             US, EU, UK) and sales platforms."

hints:
  - rule: "Must acknowledge specified certification requirements by geography"
    reasoning: "Compliance is location-dependent. A 'good quality' product is
                illegal to sell if it lacks the specific paperwork for that country."
    examples:
       - "USA: FCC (Electronics), FDA (Food/Skin), CPC (Children's Products)"
       - "Europe: CE, RoHS, REACH"
       - "Global/Retailer specific: UL, BSCI (Social Compliance)"

  - rule: "Indicate whether recommendations meet the required certifications"
    reasoning: "Suppliers often claim to have certificates they don't actually
                possess, or they provide certificates that belong to other
                factories."
    require:
       - "Request the full PDF of the Test Report, not just the Certificate cover
         page."
       - "Verify the 'Applicant' name matches the Supplier's business license name."
       - "Check that the 'Product Name/Model' on the cert matches the exact item
         being sourced."
```

## G. Source Quality Evaluation Framework

To ensure the credibility of verification results, we utilize a structured Source Quality Scoring module. The system evaluates reference URLs based on a tiered domain classification, quantifying the authority and reliability of the information sources.

### G.1. Source Tier Classification

Sources are categorized into four distinct tiers based on their domain reputation and institutional authority. Each tier is assigned a baseline score ($S_{tier}$), as detailed in Table 14.

### G.2. Scoring Methodology

The framework calculates an Overall Quality Score ($Q$) for a set of $n$ references. The scoring engine supports position-based weighting. The first cited source is treated as the primary evidence and assigned higher significance. The score is calculated as follows:

$$Q = \frac{\sum_{i=1}^{n}(S_{tier,i} \cdot w_i)}{\sum_{i=1}^{n} w_i} \tag{18}$$

Where:

- $S_{tier,i}$ is the score of the $i$-th source based on its classification.

- $w_i$ is the positional weight, defined as $w_i = \frac{1}{i}$ (e.g., 1.0, 0.5, 0.33...).

*Table 14.* Source credibility tiers.

| Tier | Category | Description & Representative Examples | Weight ($S_{tier}$) |
|------|----------|--------------------------------------|---------------------|
| **T1** | Official Platforms | First-party platforms, government (.gov) and academic (.edu) domains | 1.00 |
| | | *Examples:* `tiktok.com`, `google.com`, `microsoft.com`, `gov`, `edu` | |
| **T2** | Authoritative Data | Major data analytics, research firms and global news media | 0.75 |
| | | *Examples:* `statista.com`, `springer.com`, `wsj.com`, `economist.com` | |
| **T3** | Professional Tech | Tech-focused media, developer communities and industry databases | 0.50 |
| | | *Examples:* `techcrunch.com`, `github.com`, `stackoverflow.com` | |
| **T4** | Unknown Sources | Unrecognized domains, small individual blogs or user-generated content | 0.25 |

### G.3. Quality Grading and Recommendations

Based on the final score $Q$ and the presence of official sources, the system assigns a letter grade (A–F).

- **Grade A**: Requires $Q \geq 0.75$ and the inclusion of at least one T1 (Official) source.

- **Grade F**: Assigned if $Q < 0.3$ or if all sources are entirely unknown.

Results graded below B trigger automated recommendations, such as suggesting the addition of government data or first-party platform documentation to strengthen the claim.

## H. Case Study: Detecting Failure Modes

This section presents a detailed case study demonstrating how JADE detects failure modes missed by holistic evaluation. We utilize a specific user query regarding the market analysis for customizable cashmere coats to illustrate distinct types of verification mechanics, including real-time checks, content-dependent reasoning, and logic gating.

### H.1. Context: User Query

```
User Request:
"I would like to sell customizable cashmere coats for women.  The price range would
be in between $999 CAD and $1,199 CAD. Customers can choose from 10 colours; we
customize based on the customer's measurements.  We plan to sell to U.S., Canadian,
and UK customers.  Our primary platform would be Etsy.  These coats are made using
a premium cashmere-wool blend, with cashmere taking 50% or more of the fabric
content.  The coats are handmade.  We can also slightly adjust the style (add
pockets, belts, etc.).  We want to start small.  For the first year, a $10,000 sale
would be desirable.  Could you analyze the market to see if there is an opening?"
```

### H.2. Case A: Real-time Verification (Currency Conversion)

AGENT REPORT (EXCERPT)

```
...To position your brand effectively on Etsy, we must align with the premium tier.
Based on your desired price range, the conversion is as follows:
> $999 -- $1,199 CAD equates to approximately $730 -- $875 USD.
This positions you significantly above the $300--$400 average for wool coats...
```

JADE VERIFICATION LOG

---

**Verification Result: Currency Check**

```
item_id:    5
type:    evidence
checklist:    Verify the currency conversion that $999--$1,199 CAD equates to
approximately $730--$875 USD using a current mid-market exchange rate.
tool:    XE Currency Converter (CAD to USD, Jan 17 2026)
score:    0.015 (FAIL)
analysis:    The currency conversion is inaccurate based on the current mid-market
exchange rate.  The claim uses an exchange rate of approx 0.73 USD/CAD, whereas
the actual mid-market rate is approx 0.719--0.720 USD. This results in a material
deviation of approx 1.5%.
```

---

**Analysis Conclusion**  Traditional holistic evaluations typically fail to catch this error because the numbers (730 vs. 719) look plausible to a human reviewer. Without invoking an external tool to fetch the specific exchange rate for the specific date, the slight hallucination in financial data passes unnoticed. JADE's tool-use capability ensures that quantitative claims are verified against ground truth rather than linguistic fluency.

### H.3. Case B: Content-Dependent Reasoning (Geographic Claims)

AGENT REPORT (EXCERPT)

```
...4.  Target markets (US, Canada, UK) are the largest for luxury cashmere [Source:
researchandmarkets.com]...
```

JADE VERIFICATION LOG

---

**Verification Result: Geographic Reasoning**

```
item_id:    9
type:    reasoning (source grounding)
checklist:    Verify whether the claim that US, Canada, and UK are the largest markets
is supported by the cited data.
claimed_source:    researchandmarkets.com (North America Report)
score:    0.5 (PARTIAL PASS)
analysis:    The cited source explicitly supports the U.S. market projection but
treats "North America" as a block.  It does not provide specific ranking data for the
UK or Canada.  The inclusion of the UK is an hallucination relative to the cited North
American report.
```

---

**Analysis Conclusion**  This case exemplifies the challenge of detecting **unfaithful citations**. The agent supports its claim with a valid URL, creating a veneer of veracity that often bypasses superficial human review. JADE, however, verifies specific *content entailment* rather than mere link validity. It correctly identifies that the source—being a North American report—cannot logically ground the assertion regarding the UK market. By penalizing this over-generalization, JADE exposes how models often misappropriate valid evidence to support hallucinations.

### H.4. Case C: The "Pantone" Hallucination (Gated by Evidence)

AGENT REPORT (EXCERPT)

```
To maximize social media virality at launch, we recommend including a limited
edition 'Coral Pink' coat.
Trend Signal: 'Coral Pink' has been announced as the Pantone Color of the Year for
2026.  Launching this specific shade will allow your brand to hijack the 'Color of
the Year' hashtag traffic...
```

JADE VERIFICATION LOG

```
Verification Result: Logic Gating

item_id:    12
type:    reasoning (strategy validation)
checklist:    Evaluate the validity of the recommendation to launch 'Coral Pink' coats
specifically to capture 'Color of the Year' hashtag traffic.
dependency:    Evidence Check #5 (Fact Verification):  FAILED
  (Official 2026 Pantone Color is ''Cloud Dancer'' (White), not Coral Pink.)
status:    [GATED]
score:    0.0 (FAIL)
analysis:    The marketing strategy logically relies on the premise that 'Coral Pink'
is the trend color.  Since the upstream Evidence Check #5 confirmed this premise is
factually false (Hallucination), the derived strategy is automatically invalidated
without further evaluation.
```

**Analysis Conclusion**   This demonstrates JADE's logic gating mechanism. A holistic review might praise the strategic creativity of "trend-jacking." However, JADE explicitly models the dependency between facts and reasoning. Because the foundational fact (Evidence #5) is hallucinated, the downstream strategy is operationally useless and is explicitly penalized, preventing "smooth but wrong" advice from receiving high scores.

# I. Human Alignment Experimental Setup

To ensure the reliability and objectivity of the human evaluation, we engaged **five domain experts** to annotate the generated reports. The experimental setup regarding model selection, annotation process, and scoring aggregation is detailed below.

## I.1. Model Configuration

The evaluation corpus consists of total 180 reports generated by six diverse large language models (LLMs), selected to ensure a broad coverage of capabilities and architectural differences. The specific model identifiers used in this study are:

- `Gemini Deep Research`

- `Shopping Research`

- `Claude Opus 4.5 /w Tool Use`

- `Doubao Seed-1.6 /w Tool Use`

- `Gemini 3 pro /w Tool Use`

- `Gpt-5.2 /w Tool Use`

## I.2. Scoring Aggregation Mechanism

To mitigate the impact of individual bias and potential outliers, we applied a **trimmed mean aggregation strategy**. For each evaluated item, scores were collected from all five independent experts. We systematically excluded the single highest and single lowest scores, calculating the final ground truth score as the arithmetic mean of the **remaining three scores**. This trimming is used only to construct the aggregated ground-truth target for model-alignment analysis; it is not used when computing inter-annotator agreement.

## I.3. Inter-Annotator Agreement

We evaluated the internal consistency of the expert annotations to validate the quality of our ground truth data. We calculated the average pairwise Pearson correlation across **all five experts without trimming**. The **average inter-annotator correlation was** 0.831, demonstrating a high degree of agreement among the experts and confirming the reliability of the human evaluation baseline used in this study.

*Table 15.* Comparison of JADE with existing agent benchmarks across key dimensions.

| Benchmark | Real-world Authenticity | Temporal Sensitive | Reference-free Eval | Expert-aligned Logic |
|---|---|---|---|---|
| BrowseComp | ✗ | ✗ | ✗ | ✗ |
| DeepResearchBench | ✓ | ✗ | ✗ | ✗ |
| RigorousBench | ✓ | ✗ | ✗ | ✓ |
| HealthBench | ✓ | ✗ | ✗ | ✓ |
| DR.BENCH | ✓ | ✗ | ✗ | ✓ |
| **Ours (JADE)** | ✓ | ✓ | ✓ | ✓ |

## J. Comparison with Related Work

To situate our contribution within the broader landscape of agentic evaluation, we compare our proposed benchmark against several recent datasets: BrowseComp, DeepResearchBench, RigorousBench, HealthBench, and DR.BENCH. The comparison focuses on four critical dimensions required for evaluating professional-grade market analysis agents:

1. **Real-world Authenticity:** Whether the queries and tasks reflect genuine, complex user needs rather than synthetic or simplified prompts.

2. **Temporal Sensitive:** Whether the evaluation supports **real-time verification** of dynamic information (e.g., current exchange rates, latest policy changes) rather than relying on frozen, outdated snapshots. This addresses the *Temporal Sensitivity* challenge identified in the main text.

3. **Reference-free Evaluation:** Whether the verification mechanism supports open-ended generation without relying on a single static gold reference, addressing the *Open-endedness* challenge.

4. **Expert-aligned Logic:** Whether the evaluation criteria enforce domain-specific constraints (e.g., logical dependencies, compliance checks) rather than just surface-level factuality.

Table 15 summarizes this comparison. While recent benchmarks like RigorousBench and HealthBench have made significant strides in incorporating expert-aligned logic and real-world scenarios, they remain limited by their reliance on static reference answers. Crucially, they lack **real-time temporal awareness**, meaning they cannot verify claims against the current state of the world. In contrast, JADE utilizes tool-assisted verification to check dynamic data points at the moment of evaluation.

**JADE (Ours)** uniquely satisfies all four dimensions. Unlike BrowseComp and DeepResearchBench, which focus primarily on information retrieval recall or static QA, JADE introduces a temporal sensitive, reference-free verification framework. This allows it to robustly evaluate agents on tasks that are both hierarchically complex and dynamically changing, closing the gap between academic evaluation and real-world deployment requirements.

