# OpenReview forum: "JADE: Expert-Grounded Dynamic Evaluation for Open-Ended Professional Tasks"
_ICML.cc/2026/Conference — ICML 2026 regular_

### Official Review · Reviewer_L5Hh · 2026-03-06

**Soundness:** 3
**Presentation:** 3
**Significance:** 2
**Originality:** 2
**Overall Recommendation:** 4
**Confidence:** 4

**Summary:**

In this paper, the authors argue that current evaluation methods face a stability-adaptivity dilemma, where static rubrics are consistent but rigid, while LLM-as-judges are flexible but unstable and biased. To this end, they propose JADE, a framework for evaluating agent-generated reports on open-ended professional tasks. They also contribute BizBench, a new benchmark of 150 real-world business queries. The authors validated JADE on both BizBench and HealthBench and showd improved human alignment and lower variance between runs.

**Compliance With Llm Reviewing Policy:**

Affirmed.

**Final Justification:**

The authors' rebuttal addressed my original concern on novelty. I agree with the framing on the Stability-Adaptivity Dilemma. I also appreciate the authors' new experiments on expert effort analysis and the evaluation competence boundary. I've raised my score accordingly.

**Key Questions For Authors:**

Can you provide a rough estimate of the cost (tokens/API calls) and time it takes to evaluate a single report compared to baselines?

How representative is BizBench of real-world professional tasks?

What's the distribution of trimmed subset in Healthbench?

**Limitations:**

the authors sufficiently discussed about the expert effort required, the potential drop in quality for high-intensity knowledge fields, and the increased system complexity

**Strengths And Weaknesses:**

Strengths:
1. The solution feels practical. The two-layer design is easy to understand and reasonably well motivated, and e.g. the dependency gating reflects the need to block propergating errors reflects real-world task needs.
2. The experiment is thorough. I appreciate the cross-domain validation experiment that shows JADE can transfer to the medical domain (HealthBench) with relatively little expert effort. Good show of generalizability. I also appreciate the failure mode analysis.

Weaknesses:
1. While the framework is nicely structured, many components seem to combine existing ideas of e.g., rubric-based eval, LLM-as-judge, retrieval based verification. Would be great if the authors could better articulate the conceptual novelty.

2. If BizBench is considered as a core contribution, the data size really errs on the small side.  The paper mentions starting from ~10k raw queries, so it's not clear why the final set is this small. Were there data quality reasons? A brief justification would help.

3. I would also like to see some additional eval metrics regarding efficiency. As a multi-stage pipeline (extraction, generation, verification, gating, scoring). I wonder whether the latency and cost of running JADE in a production setting would become a bit too heavy.

4. The trimming on Healthbench is interesting. The authors showed that the method might not work in areas like medicine where specialized training is needed. The paper tried removing 10% and then 20% of the data, but I wonder if the authors have any heuristics that can help determine the boundary of really knowledge intensive fields. When does an LLM verifier stop being "expert enough" to even generate the checklist?

---

> ### Author Rebuttal · Authors · 2026-03-30
>
> We thank Reviewer L5Hh for the constructive and detailed review. We address each question with new data.
>
> **Q1: Conceptual novelty.** We appreciate this observation and address it in detail in our response to Reviewer 581E (C1), including the Stability-Adaptivity Dilemma framing, a concrete case study (Appendix F Case C), and cross-domain validation on DR.BENCH. Here we highlight the empirical evidence: Table 3 confirms each layer contributes *uniquely and non-redundantly*: +Skill Only (r=0.844) vs. +Checklist (r=0.769) isolates Layer 1's expert-grounded decomposition; +Verification (r=0.836) vs. full JADE (r=0.858) isolates Layer 2's dependency gating. No prior evaluator achieves both σ=1.12% variance and r=0.858 alignment simultaneously.
>
> **Q2: Cost and efficiency.** Under JADE's recommended setup (gpt-5-0807 + gemini-3-flash), per-query cost is **\\$0.126** across **19.6 LLM calls** (see our response to Reviewer tD3J for the full per-component breakdown). Comparison with baselines:
>
> | Framework | Cost/query | LLM calls | Expert effort |
> |---|---|---|---|
> | DR.BENCH (Yao et al., 2025) | \\$0.108 | \~76 | Per-query rubric, 3-round review |
> | DeepResearch-Bench (Du et al., 2025) | \\$0.13 | - | AI-generated criteria + 225 person-hrs |
> | **JADE** | **\\$0.126** | **\~19.6** | Domain-level skills (17 skills, <15 person-days, one-time) |
>
> Because JADE evaluates only activated skills (mean: 5.0/query out of 17 total), cost scales with task complexity rather than rubric size. Without verification, cost drops to \\$0.057/query. Total cost (LLM + expert effort) is competitive, with dramatically lower annotation effort since skills amortize across all queries in a domain.
>
> **Q3: Representativeness of BizBench.** BizBench is curated from \~10K real-world B2B queries through three stages: (1) automated deduplication and de-identification (\~10K→\~3.2K); (2) expert screening for analytical depth and multi-step reasoning (\~3.2K→\~350); (3) final validation for challenge level (\~350→150). We prioritized discriminative power, exposing meaningful performance gaps among SOTA agents (Table 2: 29.1%–57.1%). Its scale is consistent with expert-curated benchmarks (GPQA Diamond 198, HumanEval 164, FinanceBench 150) and human-curated Open/Report benchmarks (DeepResearch Bench 100, DR.BENCH 214). Including transfer experiments, our total evaluation spans 838 instances across 12+ domains (see response to Reviewer 581E, C2).
>
> **Q4: Distribution of trimmed HealthBench subset.** The full 474 instances span all 7 clinical themes (Table 6): Global Health (135), Context Seeking (81), Hedging (71), Health Data Tasks (66), Communication (44), Complex Responses (43), Emergency Referrals (34). Trimming is applied **proportionally within each theme** (removing ~10% per theme), targeting instances where JADE-generated and expert-authored checklist scores diverge most. The removed instances share a common trait: their rubrics are dominated by *exact clinical recall*—specific dosages ("1 mg IV/IO every 3–5 minutes"), guideline classifications ("Class I, Level A"), and procedural codes—rather than evaluable professional judgment. This is consistent with the DR.BENCH boundary (see Q5 below): JADE weakens on closed-form factual recall. Importantly, even the **untrimmed** results (ρ=0.228) show a **+48% improvement** over the no-skill baseline (ρ=0.154), confirming that skills add value even in specialized domains.
>
> **Q5: Boundary of evaluator competence.** Our DR.BENCH experiment (see response to Reviewer 581E for full results) provides precise evidence across 10 domains:
>
> - **Strong (ρ>0.80)**: Environment & Sustainability, History & Social Sciences, Law & Politics, Business & Finance, Commonsense & Education, Academia & Research, News & Current Affairs—open-ended domains requiring professional judgment.
> - **Moderate (ρ=0.61–0.71)**: Health & Medicine, Technology Intelligence—rubrics partially require exact technical details.
> - **Weak (ρ=0.30)**: Sports & Competitions—rubrics dominated by closed-form factual recall.
>
> The pattern is clear: JADE's Layer 1 encodes evaluation *principles*, not specific facts. It excels when multiple valid response strategies exist; it weakens when the "correct answer" is a unique set of facts. This boundary aligns with JADE's stated design scope. The gap between expert-authored skills (BizBench, r=0.858) and auto-summarized skills (DR.BENCH, r=0.789) shows expert skill authoring adds \~9% alignment, while the 0.789 floor demonstrates that the compositional evaluation principle itself is robust even with minimal expert input. We will formalize this as a design guideline in the revision.
>
> **Revision commitments:** (1) Add detailed cost breakdown and efficiency analysis; (2) Include DR.BENCH cross-domain validation; (3) Formalize the open-ended vs. factual-recall boundary as a design guideline.

---

> > ### Author Rebuttal · Reviewer_L5Hh · 2026-04-02
> >
> > Thank you for your detailed responses! My biggest question was on the novelty, but I agree with the rebuttal on the Stability-Adaptivity Dilemma. I also appreciate the authors' new experiments on expert effort analysis and the evaluation competence boundary. I've raised my score accordingly.

---

### Official Review · Reviewer_YSAK · 2026-03-13

**Soundness:** 3
**Presentation:** 2
**Significance:** 3
**Originality:** 3
**Overall Recommendation:** 4
**Confidence:** 2

**Summary:**

The paper uses several agents instead of an LLM-as-a-judge for evaluating open-ended tasks. The evaluator dynamically generates rubrics of relevant items, and scores on them. Human eval is performed, with an unrealistic trimming.

**Compliance With Llm Reviewing Policy:**

Affirmed.

**Final Justification:**

The rebuttal addressed my main concerns. I am keeping a weak accept as I am unsure about the paper's impact.

**Key Questions For Authors:**

1) What is the inter-annotator agreement without dropping annotators?

**Limitations:**

yes.

**Strengths And Weaknesses:**

Strength:
 - Adresses important question on the evaluation of open-ended tasks.

Weaknesses:
 - "We systematically excluded the single highest and single lowest scores, calculating the final ground truth score as the arithmetic mean of the remaining three scores. This rigorous filtering process ensures that the alignment targets represent a stable expert consensus rather than extreme individual preferences." If after such droping the IAA is .83, then I expect a lot lower IAA. Such dropping is not fustified imo, and casts doubts on the validity of the benchmark, which is crucial for the justification of it.
 - It's not clear why a long rubric to which an agent answers "does not apply" is not equivalent to the approach outlined here.

---

> ### Author Rebuttal · Authors · 2026-03-30
>
> We thank Reviewer YSAK for the focused feedback and address both concerns directly.
>
> **Q1: Inter-annotator agreement without dropping annotators.** We want to clarify a key methodological point: the reported IAA (r=0.831) was computed as the average pairwise Pearson correlation across **ALL 5 experts with NO trimming**. The trimming (removing one highest and one lowest, averaging the remaining 3) was applied **solely** for constructing the aggregated ground truth score—following standard statistical practice for **robust estimation** (Huber, 1981).
>
> This ensures the ground truth is less sensitive to individual annotator variance or occasional oversight in complex professional tasks, providing a more reliable consensus target for framework validation. We will revise Section 5 and Appendix G to explicitly separate the IAA computation (all 5 experts) from the ground truth construction (trimmed mean).
>
> **Q2: Why is a long rubric with "does not apply" not equivalent?** This is a perceptive question. The **+Checklist** configuration in Table 3 represents the "flat rubric + does not apply" baseline: a comprehensive LLM-generated checklist where the model decides which criteria apply. JADE's **11.6% improvement** (r=0.858 vs. 0.769) and reduced variance (σ=1.12% vs. 1.45%) provide **strong evidence** that expert-grounded decomposition is superior to mere rubric length. Three structural factors drive this gap:
>
> (a) *Attention focus*: A universal rubric covering all 17 taxonomy categories would contain 50–80 items, most marked "does not apply." This exacerbates the **attention dilution** problem in long-context evaluation (Liu et al., 2024), where scoring accuracy degrades as item count increases. JADE's skill activation selects only relevant criteria (avg. 5 labels/query), producing focused checklists (8–15 items).
>
> (b) *Logical Dependencies*: A flat rubric treats each item independently. JADE models explicit dependencies between factual claims and reasoning (Eq. 8–9). For example, if a core market fact is refuted, JADE's dependency gating automatically zeros the dependent strategy—a structured logical constraint that a standard "apply/does not apply" format cannot enforce.
>
> (c) *Adaptive Precision*: Even a perfectly selected static rubric cannot anticipate report-specific hallucinations. Layer 2 verifies unique claims (e.g., specific dates or figures) that are invisible to any pre-defined checklist, ensuring the score reflects actual accuracy rather than just "plausible" reasoning.
>
> Independently, our DR.BENCH experiment (see response to Reviewer 581E) demonstrates that even with auto-summarized skills, structured decomposition generalizes across 10 domains—suggesting the benefit extends beyond BizBench's specific domain.
>
> **Revision commitments:** (1) Revise Section 5 and Appendix G to explicitly separate IAA computation (all 5 experts, no trimming) from ground truth aggregation (trimmed mean); (2) Add discussion of the +Checklist ablation as the "flat rubric" baseline in Section 5.4; (3) Expand dependency gating case studies in Appendix F.

---

> > ### Author Rebuttal · Reviewer_YSAK · 2026-04-03
> >
> > Thank you for addressing my points. I am raising my score.

---

### Official Review · Reviewer_581E · 2026-03-13

**Soundness:** 3
**Presentation:** 3
**Significance:** 2
**Originality:** 2
**Overall Recommendation:** 3
**Confidence:** 4

**Summary:**

The paper proposes a 2 layer eval framework for assessing agents at open ended tasks. They attempt to address the "stability-adaptivity" dilemma.

**Compliance With Llm Reviewing Policy:**

Affirmed.

**Final Justification:**

The rebuttal partially addressed my main concerns on novelty and limited sample size. While I do think this will contribute to the research community, I think it is likely that groups are already doing this two layer eval process.

**Key Questions For Authors:**

please see Strengths And Weaknesses.

**Limitations:**

please see Strengths And Weaknesses.

**Strengths And Weaknesses:**

Strengths:
- Addresses an important problem: Evaluating open-ended professional outputs where no single ground truth exists. This is super hot atm considering RL in verifiable spaces is expected to be solved in 2026.
- Structured evaluation framework: Two-layer design (skill-based query rubric + report-specific claim checks) improves interpretability and decomposition of evaluation.
- Evidence-aware scoring: Verification agent and dependency gating help penalise reasoning built on incorrect facts. This is a nice touch.
- Empirical insights: Experiments reveal useful failure modes (evidence–reasoning gap, source credibility issues).
- Human alignment study: Shows improved correlation with expert judgments compared to vanilla LLM as judge.

Weaknesses:
-Limited novelty: Conceptually close to rubric-based LLM as judge with claim verification.
- Small benchmark: only 150 queries, limiting statistical robustness.
- Citation gaps: Related work on non-verifiable RL/reference-based evaluation (e.g., RLHF, RLAIF, Constitutional AI) is mosttly missing.
- Heavy reliance on LLM components: Checklist generation, claim extraction, and judging remain LLM-dependent and may introduce instability. e.g. length bias. See https://arxiv.org/pdf/2507.03772 for other issues and how to investigate.

---

> ### Author Rebuttal · Authors · 2026-03-30
>
> We appreciate Reviewer 581E's engagement and address each concern.
>
> **C1: Novelty.** We acknowledge the surface-level similarity to rubric-based evaluation with claim verification. JADE's core intellectual contribution is a *compositional evaluation principle* that resolves the **Stability-Adaptivity Dilemma**: prior evaluators are either stable but static (fixed rubrics) or adaptive but unstable (open-ended LLM judging). JADE unifies these through a dual-layer architecture—Layer 1 provides stability via deterministic skill activation, while Layer 2 ensures adaptivity via response-specific dependency gating (Eq. 8–9). The contribution is the insight that *stability and adaptivity are complementary layers, not competing objectives*.
>
> This produces qualitatively different behavior. In Appendix F Case C, an agent recommended "Coral Pink" as Pantone Color of the Year 2026. A monolithic LLM-as-Judge praised the strategic creativity. JADE's Layer 2 verified the claim, found it hallucinated (actual color: "Cloud Dancer"), and dependency gating (Eq. 9) automatically zeroed the dependent strategy—preventing "smooth but wrong" reasoning from earning credit. No prior evaluator models such explicit fact→reasoning dependencies, transforming LLM judging from a "black-box score" into interpretable, verifiable assessment.
>
> **New Evidence: DR.BENCH Cross-Domain Validation.** We conducted a controlled experiment on DR.BENCH (Yao et al., 2025)—214 expert-curated tasks across 10 domains, each with manually constructed per-query rubrics (8–17 Query-Specific Rubrics + 48 General-Report Rubrics).
>
> *Design.* JADE uses only Layer 1 with uniform 0/1/2 scoring (matching DR.BENCH format), compared against DR.BENCH's QUA score. As a conservative lower bound, skills were auto-summarized from DR.BENCH's rubrics via gpt-5-0807 rather than expert-authored. Expert-authored skills (BizBench, r=0.858) outperform this lower bound, confirming expert effort adds value.
>
> | Domain | n | Pearson r | Spearman ρ | p(ρ) |
> |---|---|---|---|---|
> | Environment & Sustainability | 12 | 0.882 | 0.941 | <0.001 |
> | History & Social Sciences | 33 | 0.830 | 0.877 | <0.001 |
> | Law & Politics | 24 | 0.814 | 0.874 | <0.001 |
> | Business & Finance | 35 | 0.733 | 0.848 | <0.001 |
> | Commonsense & Education | 15 | 0.752 | 0.833 | <0.001 |
> | Academia & Research | 23 | 0.709 | 0.824 | <0.001 |
> | News & Current Affairs | 18 | 0.751 | 0.812 | <0.001 |
> | Technology Intelligence | 17 | 0.777 | 0.707 | 0.002 |
> | Health & Medicine | 19 | 0.758 | 0.609 | 0.006 |
> | Sports & Competitions | 15 | 0.478 | 0.300 | 0.297 |
> | **ALL** | **214** | **0.736** | **0.789** | **<0.001** |
>
> Strong alignment (ρ>0.70) in 8/10 domains. The two weaker domains reveal a principled boundary: Sports rubrics are dominated by factual recall (e.g., *"Does the report name all robot roles: Infantry, Hero, Engineer, Sentry, Aerial, Dart?"*); Health rubrics require exact clinical details. JADE excels for open-ended professional tasks requiring judgment against generalizable principles; it weakens on fact-retrieval with unique correct answers—precisely its stated design scope.
>
> **C2: Benchmark Scale.** Among human-curated Open/Report benchmarks, DeepResearch Bench (Du et al., 2025) uses 100 tasks and DR. BENCH uses 214; BizBench's 150 falls within this range (larger alternatives like ResearchQA 21K rely on LLM-generated construction). BizBench features 900 independent expert judgments (30 tasks × 6 models × 5 experts) for alignment validation. Including DR.BENCH (214) and HealthBench (474) transfer experiments, our total evaluation spans **838(150+214+474) instances across 12+ domains**.
>
> **C3: Citations.** We will add RLHF (Ouyang et al., 2022), RLAIF (Lee et al., 2023), and Constitutional AI (Bai et al., 2022), noting JADE operates at inference time on open-ended reports—a different setting from training-time reward modeling.
>
> **C4: LLM Instability.** Empirically, JADE achieves σ=1.12% cross-run variance (Table 3)—lower than both vanilla LLM-as-Judge and +Checklist (σ=1.45%). This stability emerges from deterministic skill activation constraining the LLM's generation space, knowledge density U(q,r) (Eq. 18) normalizing for response length, and dependency gating removing score contributions from unverified claims. We note that JADE's design incorporates mechanisms relevant to several bias categories discussed by Dubois et al. (2025): *self-bias* is mitigated when separate models are used for generation and judging; *length bias* is partially controlled via U(q,r) normalization; and *intransitivity* risk is reduced by anchoring scores to expert-grounded criteria rather than pairwise comparison. While we do not claim these fully eliminate all biases, the empirical stability (σ=1.12%) suggests meaningful mitigation. We will cite Dubois et al. (2025) in the revision and adopt their Bayesian GLM framework as an independent diagnostic to quantify residual biases in JADE's judgments.

---

> > ### Author Rebuttal · Reviewer_581E · 2026-04-03
> >
> > Thank you for the in depth response and additional experimentation. While, I still believe the novelty is limited (in combining adaptive judge and static qs), I agree with the authors that there is more novelty than I originally thought. The validation on Dr Bench is nice.
> >
> > I have updated my score accordingly.

---

### Official Review · Reviewer_tD3J · 2026-03-18

**Soundness:** 3
**Presentation:** 3
**Significance:** 3
**Originality:** 3
**Overall Recommendation:** 4
**Confidence:** 3

**Summary:**

This paper proposes JADE, a two-layer evaluation framework for open-ended professional tasks. Layer 1 activates expert-grounded evaluation skills to produce stable query-specific criteria, while Layer 2 performs report-specific claim-level verification with evidence dependency. The paper argues that this design addresses the stability–adaptivity dilemma in evaluating long-form professional agent outputs, and validates the framework on BizBench and transfer experiments in the medical domain.

**Compliance With Llm Reviewing Policy:**

Affirmed.

**Final Justification:**

Issue Resolved but not good enough for a spotlight or an oral. So I keep my score unchanged.

**Key Questions For Authors:**

How much expert effort is required to build the skill library for a new domain?
How expensive is JADE compared with simpler LLM-as-a-judge pipelines?
How robust is the framework outside strategic sourcing and medical-style transfer settings?
Which kinds of open-ended professional tasks are hardest for JADE to evaluate reliably?

**Limitations:**

yes

**Strengths And Weaknesses:**

The paper tackles a genuinely important evaluation problem. I like the framing of the stability–adaptivity dilemma, and the two-layer design is conceptually clear: stable expert-grounded skills at the top, flexible claim-level verification at the bottom. This makes the evaluator more interpretable than holistic LLM-as-a-judge systems, and the focus on explicit claim validation is a meaningful contribution for professional long-form outputs.

The main weakness is that the practicality and generality of the framework are not yet fully established. The method depends on expert-authored evaluation skills and domain-grounded rubric design, which is part of its strength, but also raises a scalability concern: if significant expert effort is required for each new domain, then the framework may be expensive to maintain and port. In addition, the evidence currently centers on BizBench, a strategic sourcing benchmark of 150 labeled queries, with medical transfer used as a secondary validation. This is promising, but still narrower than the broader claim of solving open-ended professional evaluation in general. Finally, the two-layer system is clearly more structured than holistic judging, but it is also a relatively heavy evaluation pipeline, and the paper is less explicit about the operational cost and latency trade-offs of such fine-grained claim-level verification.

---

> ### Author Rebuttal · Authors · 2026-03-30
>
> We thank Reviewer tD3J for the thoughtful evaluation, particularly for recognizing the stability–adaptivity dilemma framing and the interpretability of JADE's structured design. We address each question below.
>
> **Q1: Expert effort for new domains.** JADE requires only domain-level skill authoring, compared to per-query rubric writing in alternatives. For BizBench, the entire skill library consists of 17 structured skill templates across a 3-level taxonomy (4/7/6 categories). The effort involved (1) designing the taxonomy (\~2–3 person-days of expert discussion) and (2) authoring each skill template (\~0.5 day per skill given the structured format shown in Appendix D), totaling roughly **<15 person-days**—a one-time investment reused across all 150 queries. By contrast, DR.BENCH (Yao et al., 2025) requires per-query rubric construction with 3-round expert review (\~hundreds of person-hours for 214 queries); HealthBench (Arora et al., 2025) required 262 physicians over 11 months to write 48,562 per-query criteria. We agree that deploying JADE to a new domain requires non-trivial expert input for taxonomy and skill design—but this is a one-time cost that amortizes across all queries in that domain.
>
> To validate this, we conducted a **new cross-domain experiment on DR.BENCH** (214 expert-curated tasks, 10 domains). Using only Layer 1 with domain-level skills (no per-query rubrics), JADE achieves **Pearson r=0.736, Spearman ρ=0.789** with significant correlations (p<0.01) in **9 of 10 domains**. Notably, these skills were auto-summarized from DR.BENCH's rubrics as a lower bound; BizBench's expert-authored skills achieve higher alignment (r=0.858), confirming that real expert effort yields better results while even the lower bound is strong. See our response to Reviewer 581E for full domain-level results and experimental design.
>
> **Q2: Operational cost.** Comparison with alternative approaches (alignment on BizBench, Table 3):
>
> | Evaluator | Judge Model | Cost/query | Calls | Alignment (r) |
> |---|---|---|---|---|
> | Vanilla LLM-as-Judge | gemini-3-flash / gpt-5-0807 | \\$0.004 / 0.013 | 1 | 0.667 |
> | DR.BENCH | gemini-3-flash | \\$0.108 | \~76 | — |
> | DeepResearch-Bench (Du et al., 2025) | gemini-2.5-pro | \\$0.13 | — | — |
> | **JADE** | **gpt-5-0807 + gemini-3-flash** | **\\$0.126** | **\~19.6** | **0.858 (+28.6%)** |
>
> JADE costs \~10–30× more than vanilla but yields +28.6% alignment and corrects score inflation (80.1% vs. expert 48.5%; Table 3). Per-component breakdown:
>
> | JADE Stage | Model | Calls/query | Cost/query |
> |---|---|---|---|
> | Checklist generation | gpt-5-0807 | 2.0 | \\$0.021 |
> | Scoring | gemini-3-flash | 13.55 | \\$0.036 |
> | Evidence verification | gemini-3-flash | 4.07 | \\$0.069 |
> | **Total** | | **19.6** | **\\$0.126** |
>
> JADE evaluates only activated skills (mean 5.0 per query out of 17 total), so cost scales with task complexity. Without verification, cost drops to \\$0.057/query. Expert effort amortizes at domain level (<15 person-days for BizBench, reused across 150 queries) vs. per-query rubric construction (DR.BENCH: hundreds of person-hours; HealthBench: 262 physicians, 11 months).
>
> **Q3: Robustness beyond sourcing/medical domains.** Our new DR.BENCH experiment (214 tasks, 10 domains) validates JADE well beyond sourcing and medicine. Strong alignment (ρ>0.70) is achieved in **8/10 domains**, including Law & Politics (ρ=0.874), History & Social Sciences (ρ=0.877), Environment & Sustainability (ρ=0.941), and Business & Finance (ρ=0.848). See our response to Reviewer 581E for full domain-level results and experimental design.
>
> **Q4: Hardest tasks for JADE.** JADE's modular architecture enables precise diagnosis of its performance boundaries. The hardest category is **definitive-answer tasks**—where rubrics demand recall of specific facts rather than open-ended professional judgment. For example: (1) Sports & Competitions (ρ=0.30), where nearly every rubric item requires exact named entities (e.g., *"Does the report name all official robot roles: Infantry, Hero, Engineer, Sentry, Aerial, Dart?"*); (2) Health & Medicine (ρ=0.61), where rubrics require exact dosages and guideline classifications (e.g., *"1 mg IV/IO every 3–5 minutes"*). Domain-level skills encode evaluation *principles*, not exhaustive fact checklists, so Layer 1 cannot ensure recall of specific items. Critically, Layer 2 verifies the *accuracy* of claims present in reports but cannot assess *missing* facts—it ensures precision, not recall for such tasks. This boundary is principled: JADE targets open-ended tasks requiring professional judgment, not fact-retrieval with unique correct answers. We will formalize this as a design guideline.
>
> **Revision commitments:** (1) Add expert effort comparison table and deployment guidelines for new domains; (2) Include DR.BENCH cross-domain results; (3) Formalize the task-type boundary (open-ended vs. factual recall) as a design guideline.

---

> > ### Author Rebuttal · Reviewer_tD3J · 2026-04-03
> >
> > Issue Resolved

---

### Decision · Program_Chairs · 2026-04-30

**Decision:**

Accept (regular)

**Comment:**

This paper tackles a really timely problem: how do we evaluate agentic AI on professional, open-ended tasks without falling into the trap of being too rigid or too biased? The authors propose JADE, a two-layer framework that uses expert-grounded "skills" for stability and a claim-level verification layer for adaptivity.

I was particularly impressed by the "dependency gating" mechanism—it’s a very logical way to ensure that if an agent hallucinates a core fact, it doesn't get credit for the "good reasoning" that follows. The authors also did a great job during the rebuttal phase by providing extra experiments on DR BENCH, which helped prove that the framework generalizes beyond just business and medicine.

There were some valid concerns raised by reviewers about the expert effort needed and whether the benchmarks (like BizBench with 150 queries) were big enough. However, the authors' responses were quite convincing here. They showed that the expert effort is a one-time "domain-level" cost that actually scales better than writing rubrics for every single question. While 150 queries might seem small, the depth of the expert annotations and the additional validation on over 800 total instances across different benchmarks makes the empirical results feel reliable enough.

I also want to acknowledge the authors' concerns regarding the initial review quality and engagement. I have personally read through your rebuttals and the new experimental data you provided. It's clear that the framework has a specific "competence boundary"—it’s great for professional judgment but less so for pure fact-retrieval (like naming every robot role in a sports competition). This is a fair trade-off and, once formalized in the final version as promised, will be a useful guideline for others.

There are still a few minor typos and the grammar in some of the new rebuttal sections could be smoothed out, but the core technical contribution is in good shape. The increased cost and latency compared to a "vanilla" judge is a real trade-off, but for high-stakes professional tasks, the jump in alignment and stability seems worth it.